# M$^2$-Miner: Multi-Agent Enhanced MCTS for Mobile GUI Agent Data Mining

**Rui Lv**[1,*], **Juncheng Mo**[1,2,*], **Tianyi Chu**[2], **Chen Rao**[1,2], **Hongyi Jing**[1], **Jiajie Teng**[1],
**Jiafu Chen**[1,2], **Shiqi Zhang**[1], **Liangzi Ding**[1], **Shuo Fang**[1], **Huaizhong Lin**[2],
**Ziqiang Dang**[1,✉], **Chenguang Ma**[1,✉], **Lei Zhao**[2,✉]

[1]Ant Group,    [2]Zhejiang University

ZiqDangQaQ@gmail.com; chenguang.mcg@antgroup.com; cszhl@zju.edu.cn

## Abstract

Graphical User Interface (GUI) agent is pivotal to advancing intelligent human-computer interaction paradigms. Constructing powerful GUI agents necessitates the large-scale annotation of high-quality user-behavior trajectory data (*i.e.*, intent-trajectory pairs) for training. However, manual annotation methods and current GUI agent data mining approaches typically face three critical challenges: high construction cost, poor data quality, and low data richness. To address these issues, we propose M$^2$-Miner, the first low-cost and automated mobile GUI agent data-mining framework based on Monte Carlo Tree Search (MCTS). For better data mining efficiency and quality, we present a collaborative multi-agent framework, comprising InferAgent, OrchestraAgent, and JudgeAgent for guidance, acceleration, and evaluation. To further enhance the efficiency of mining and enrich intent diversity, we design an intent recycling strategy to extract extra valuable interaction trajectories. Additionally, a progressive model-in-the-loop training strategy is introduced to improve the success rate of data mining. Extensive experiments have demonstrated that the GUI agent fine-tuned using our mined data achieves state-of-the-art performance on several commonly used mobile GUI benchmarks. Our work will be released to facilitate the community research.

## 1 Introduction

GUI agents, which operate software applications by understanding user intent and executing sequences of actions within graphical user interfaces, have emerged as a promising direction in both academia and industry. Powered by recent advances in MLLM, these agents are envisioned to automate a wide range of tasks, from navigating mobile apps to controlling complex desktop environments. Currently, GUI agents still face a critical limitation: their performance heavily relies on high-quality intent-to-trajectory data for training.

Existing GUI agent datasets typically rely on either manual intent-trajectory annotation Rawles et al. (2023); Li et al. (2024); Zhang et al. (2025b); Deng et al. (2023) or naive automated mining method Sun et al. (2025); Putta et al. (2024) to acquire prior knowledge about specific applications. Most of these datasets adopt an intent-to-flat-trajectory structure, as shown in Fig. 1(a), which means they only record a single successful path for each intent without capturing the full exploration process. Such an incomplete structure will lead to low richness of intent and be insufficient to support the training of a robust GUI agent model. We summarize the major limitations of current GUI agent data construction approaches as follows: 1) **high construction cost**: existing manually annotated GUI agent data typically requires several hours to create each entry; 2) **poor data quality**: existing datasets (including both manually labeled and automatically mined data) often contain redundant steps, ambiguous intent descriptions, and biased operation paths; and 3) **low data richness**: existing datasets suffer from monotonous intents and only record the actions for each step, lacking other descriptive information.

---

*  Rui Lv and Juncheng Mo contributed equally to this work.
✉ Corresponding authors: Ziqiang Dang, Chenguang Ma and Lei Zhao.

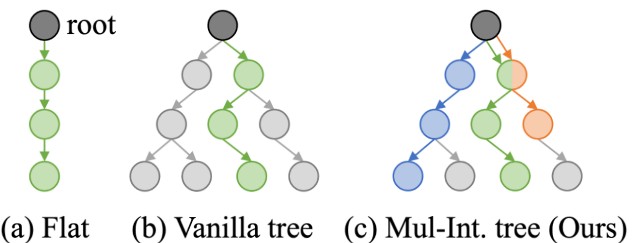 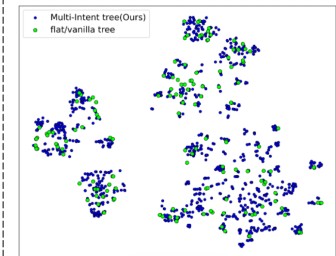

(a) Flat    (b) Vanilla tree    (c) Mul-Int. tree (Ours)

Figure 1: **Left:** Different GUI data structure. (a) Flat human-annotated GUI data, which stores only the single trajectory corresponding to each intent. (b) Vanilla tree-structured GUI agent data, where each tree is associated with a single intent. (c) Our multi-intent tree structure, where each tree contains multiple intents and their corresponding trajectories. **Right:** t-SNE Visualization of intent distributions. Compared to existing methods, our intent recycling strategy produces a more diverse set of intents from the same initial intent.

To address the above challenges, we first introduce the Monte Carlo Tree Search (MCTS) Kocsis & Szepesvári (2006) algorithm into mobile GUI agent data mining task and propose a tree-structured representation to record the complete mining process. It is noted that directly applying vanilla MCTS to GUI agent data mining will result in low efficiency due to random expansion and rollout-based reward calculation. Therefore, we present a **collaborative multi-agent framework** composed of InferAgent, OrchestraAgent, and JudgeAgent. Specifically, InferAgent improves expansion efficiency by increasing the hit rate of correct actions corresponding to a given intent. OrchestraAgent merges equivalent actions and ranks them by confidence, further boosting data mining efficiency. JudgeAgent replaces result-based rewards with process-based rewards, significantly reducing the computational cost of simulation phase. As a result, our collaborative multi-agent framework achieves exponential improvements in mining efficiency. Besides, motivated by the observation that non-primary paths in the tree hold the potential to serve as valuable intent-trajectory data, we propose a novel **intent recycling strategy** to extract these extra interaction trajectories, which improves mining efficiency and enriches intent diversity. Such strategy evolves the vanilla tree representation (Fig. 1(b)) of prior works Putta et al. (2024) into the augmented structure depicted in Fig. 1(c). Specifically, upon completion of the mining process, the recycling process begins by re-evaluating all paths in the tree with a dedicated intent recycling filter to identify suitable paths. Subsequently, new intents are generated for these identified trajectories, concluding the recycling phase. To intuitively show the enhanced diversity, we provide a comparative visualization of intent distributions in the right part of Fig. 1. The visualization is achieved by reducing the dimensionality of the intent text embeddings (obtained via Sentence Transformer Reimers & Gurevych (2019)) to two dimensions using t-SNE Maaten & Hinton (2008). Furthermore, we introduce a **progressive model-in-the-loop training strategy** to improve mining success rate and generalization in new GUI environments, through which agents capabilities and data complexity grow in tandem, resulting in significant performance improvements, especially in unseen scenarios.

Extensive experiments demonstrate that GUI agents trained on data from $M^2$-Miner achieve SOTA performance. Moreover, we conduct comprehensive ablation studies to validate the effectiveness of each key component: our collaborative multi-agent framework, the intent recycling strategy, and the progressive training strategy. Compared with vanilla MCTS, our method achieves higher mining efficiency (*e.g.*, $64\times$ boost at task length 9) and success rate, while also generating more diverse intents. Furthermore, our experimental results show that the mined data enriched with descriptions provides greater benefits for GUI agent training than traditional flat datasets. Our work offers a valuable data foundation and mining paradigm to support the advancement of the GUI community.

The main contributions of this paper can be summarized as follows: **(1)** We propose $M^2$-Miner, the first automated mobile GUI agent data-mining framework based on MCTS, which incorporates a collaborative multi-agent framework consisting of InferAgent, OrchestraAgent, and JudgeAgent, to jointly enhance mining efficiency and data quality. **(2)** To enhance intent richness and further improve mining efficiency, a novel intent recycling strategy is introduced for extracting additional valuable trajectories. **(3)** To improve the mining success rate and enhance generalization in unseen environments, we present a progressive model-in-the-loop training strategy. **(4)** Extensive experiments demonstrate the superiority of our approach in mining high-quality and rich intent trajectory data. Specifically, GUI agents trained on our mined data achieve SOTA performance.

## 2 RELATED WORKS

**GUI Agents.** With the rapid development of MLLMs, building intelligent GUI agents with automated operation capabilities for mobile devices has emerged as a leading research trend. Represented by OmniParser Lu et al. (2024), AppAgent Zhang et al. (2025a), Mobile-Agent Wang et al. (2024) and WebVoyager He et al. (2024), prompt-based GUI agents leverage large multimodal models to perform actions without explicit training. Mobile-Agent employs GPT-4V Yang et al. (2023) with visual perception modules to interpret screenshots and generate operations via prompts. Similarly, WebVoyager uses a ReAct-style Yao et al. (2023) prompting mechanism to annotate webpage elements via bounding boxes and plan click or navigation steps end-to-end. While these prompt-only approaches reduce the need for training, they are prone to hallucinations because the utilized large models lack concrete prior of GUI environments. Subsequent works generally conduct fine-tuning on application-specific trajectories or integrate trainable agent modules Cheng et al. (2024); You et al. (2024); Liu et al. (2024); Qin et al. (2025); Wu et al. (2025b); Ye et al. (2025); Lu et al. (2025b); Xu et al. (2025a). SeeClick Cheng et al. (2024) relies on vision-based grounding through pre-training on ScreenSpot screenshots and shows improved task execution. UI-TARS Qin et al. (2025) takes this further with end-to-end training on screenshot-action pairs, leveraging iterative online trace collection to achieve SOTA results across multiple benchmarks. AutoGLM Liu et al. (2024) propose a foundation GUI agent which combines planning and grounding strategies, applying online curriculum reinforcement learning to progressively improve performance in browser and mobile domains. These methods suffer from limited long-sequence reasoning, poor exception handling, and weak generalization. The key factor behind these issues is the lack of sufficient high-quality GUI data.

**GUI Agent Data Production.** The training of GUI agents heavily relies on intent–trajectory data. Existing GUI agent datasets are entirely manually constructed, such as AITW Rawles et al. (2023), AITZ Zhang et al. (2024), AndroidControl Li et al. (2024), GUI-Odyssey Lu et al. (2025a), and AMEX Chai et al. (2025). This manual annotation process is very costly, demanding specialized annotation tools and experienced annotators. Recently, several studies have begun to utilize automated data mining techniques to efficiently collect intent–trajectory data. GUI agent data mining refers to the process of collecting and labeling interaction trajectories using an automated framework. Early data mining studies were primarily designed to support LLM-based GUI agents operating in web environments, where the agents perceive the interface through the HTML code. For example, AgentQ Putta et al. (2024) pioneers a data mining approach leveraging Large Language Models for intent–trajectory mining in web scenarios. However, its applicability is limited to parsable web environments, as it lacks support for mobile scenarios that demand rich multimodal perception. Furthermore, the method suffers from low efficiency due to the inherent limitations of native MCTS. With the rapid advancements in VLMs, a substantial amount of recent GUI agent research has shifted towards purely vision-based approaches. However, research on data mining methods tailored to such vision-based GUI agents is still largely under-explored. A representative work in this line of research is OS-Genesis Sun et al. (2025), which employs a fully automated, interaction-driven GUI data processing pipeline without reliance on predefined tasks or manual annotations. Initially, the agent performs rule-based, step-wise interactions to explore the GUI environment, collecting raw state–action pairs. Then, through a process of inverse task synthesis, these low-level interactions are transformed into action instructions and task intents, thereby generating intent–trajectory pairs in an unsupervised manner. Mobile-Agent-v3 Ye et al. (2025) presents a self-evolving pipeline for generating intent–trajectory data. It combines high-quality intent generation, automated interaction, trajectory validation, and challenging-task guidance within a reinforcement fine-tuning loop, steadily improving the agent's success rate. To the best of our knowledge, $M^2$-Miner is the first automated mobile GUI agent data-mining framework based on MCTS.

## 3 METHOD

Given the urgent demand for high-quality GUI agent data and the laborious challenge of manually annotating interaction trajectories, we propose $M^2$-Miner, the first automated mobile GUI agent data-mining framework based on Monte Carlo Tree Search (MCTS), as illustrated in Fig. 2. The preliminaries of MCTS are detailed in Appendix A.1. Based on vanilla MCTS, we specifically employ a collaborative multi-agent framework, which comprises an InferAgent, OrchestraAgent,

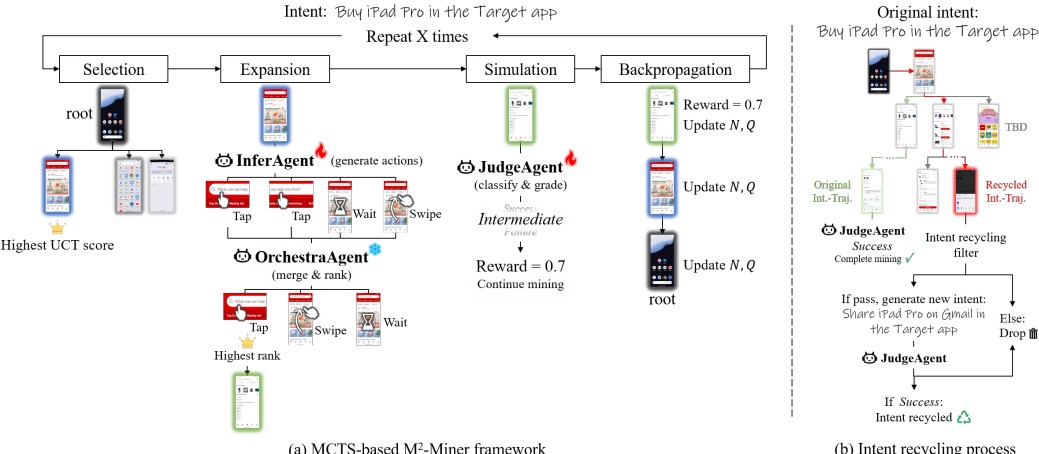

Figure 2: **Overview**. (a) The mining process of $M^2$-Miner consists of four phases: Selection, Expansion, Simulation, and Backpropagation. During these phases, InferAgent, OrchestraAgent, and JudgeAgent are employed for exploration guidance, acceleration, and state evaluation. (b) Intent recycling process. For a selected node, its corresponding trajectory is first passed through a dedicated intent-recycling filter, then a novel intent is generated via MLLM. If the JudgeAgent verifies that the generated intent aligns with the trajectory, the recycling is considered complete.

and JudgeAgent, to enhance the expansion and simulation phases. Our approach reduces costs and improves both efficiency and mining success rate. Furthermore, we propose an intent recycling strategy to fully extract valid trajectories from the intent trajectory trees. Finally, to further boost the mining success rate, we introduce a model-in-the-loop training strategy to continuously evolve the capabilities of these agents. In the rest of this chapter, Sec. 3.1 first details the formulation of intent-trajectory tree; Sec. 3.2 thoroughly describes the collaborative multi-agent architecture; Sec. 3.3 presents our novel intent recycling strategy; and Sec. 3.4 introduces our model-in-the-loop training strategy.

## 3.1 INTENT-TRAJECTORY TREE FORMULATION

Within the Monte Carlo Tree Search framework, our method constructs *intent-trajectory trees* to efficiently mine GUI intent trajectories within mobile applications.

Specifically, the intent-trajectory tree can be formalized as $\mathcal{T} = (\mathcal{V}, \mathcal{A}, \mathcal{P}, \mathcal{I})$ where $\mathcal{V}$ is the set of nodes in the tree, each node $v \in \mathcal{V}$ corresponds to a state $s$ in the environment, $\mathcal{A}$ represents the set of executable GUI actions (*e.g.*, click, swipe, and type), with the available actions at state $s$ denoted as $\mathcal{A}(s)$, $\mathcal{I}$ denotes the set of user intents, typically specified as natural language task descriptions, and $\mathcal{P}$ represents the set of edges, capturing the parent-child relationships in the search tree, $(s, a, s') \in \mathcal{P}$ if action $a$ transitions state $s$ to $s'$.

Each node $v$ in the search tree is defined as a tuple $(img_v, meta_v, Q_v, N_v, stat_v)$, where $img_v$ denotes the screenshot of the environment at node $v$, $meta_v$ is the description of the action that led to node $v$, $Q_v$ represents an aggregated value estimate of node $v$ (*e.g.*, likelihood of task success), $N_v$ is visit count of node $v$, utilized for balancing exploration and exploitation during search, and $stat_v$ refers to the task completion status for node $v$, with possible values of *success*, *failure*, or *intermediate*.

Based on such a tree formulation, given an initial intent $I_0$ and a starting GUI state $s_0$, our objective is to efficiently construct a tree that contains a path corresponding to a valid interaction trajectory $\tau = (s_0, a_0, s_1, \ldots)$ satisfying $I_0$.

## 3.2 COLLABORATIVE MULTI-AGENT FRAMEWORK

When vanilla MCTS is directly applied to GUI agent data mining, the mining efficiency and success rate are severely impacted by the expansion and simulation phases. Specifically, the expansion phase randomly selects a child node to expand. However, due to the complexity of GUI agent tasks

and the immense exploration space, this approach leads to a large number of invalid and failed explorations, which in turn significantly increases mining costs and dramatically lowers the mining success rate. Concurrently, scoring the current node during the simulation phase based on future exploration results leads to a slow mining speed and low efficiency. To address these issues, we propose a collaborative multi-agent framework, which comprises three key agents (InferAgent, OrchestraAgent, and JudgeAgent). These three agents enhance the expansion and simulation phases of canonical MCTS. Fig. 2(a) illustrates the flowchart of the MCTS algorithm integrated with our multi-agent framework.

During the selection phase, the algorithm traverses the tree and selects a node to expand based on the UCT score. This phase does not involve any agents. Subsequently, in the expansion phase, InferAgent is responsible for inferring the actions most likely to achieve the target intent. Specifically, the InferAgent generates $K$ candidate actions for the selected node $v$ based on the GUI screenshot of node $v$. To prevent redundant generation, previously produced actions will be incorporated into the prompt when generating a new action. Besides, to ensure action space diversity, we leverage multiple distinct MLLMs to generate $K$ actions. Fig. 2(a) shows an example where two tap actions, one wait, and one swipe are generated. Note that the two tap actions are essentially equivalent. Next, OrchestraAgent merges equivalent actions and ranks the remaining actions based on their likelihood of achieving the target intent, which avoids redundant expansions and efficiently guides the search toward promising trajectories. Specifically, the OrchestraAgent selects the most promising action at each iteration via a multi-choice question approach, ultimately yielding a sorted action queue through $K-1$ queries. Subsequently, the sorted actions are utilized to generate new child nodes, which are then assigned decreasing initial UCT values based on the actions' order in the queue. As depicted in Fig. 2(a), the tap action is ultimately ranked first. This action is then executed, causing the virtual machine to transition to a new page. Consequently, the corresponding node is expanded in the search tree. The mining process then proceeds to the simulation phase. In this stage, the JudgeAgent analyzes the GUI screenshot of the newly expanded node, determines the task completion status of the node (*i.e.*, $stat$) and calculates the rewards to update the $Q$-value of the node. When a node's status is terminal (*i.e.*, success or failure), the reward is set to 1 or 0, respectively. For intermediate nodes, we train the JudgeAgent to predict their rewards for achieving the target intents, outputting only *"valid"* or *"invalid"*. Based on this paradigm, we utilize the logits from the MLLM head corresponding to the token *"valid"* as the raw reward for intermediate nodes. To improve stability and comparability across different states, we introduce a normalization process using a softmax function to transform logits into a probability in $[0, 1]$. The normalized reward for intermediate nodes is thus computed as:

$$r_{\text{intermediate}} = \frac{\exp\left(logits_{\text{valid}}\right)}{\exp\left(logits_{\text{valid}}\right) + \exp\left(logits_{\text{invalid}}\right)}, \tag{1}$$

where $logits_{\text{valid}}$ and $logits_{\text{invalid}}$ denote the two output logits corresponding to the respective tokens. This normalization ensures that rewards reflect the relative confidence of the model in a bounded and interpretable manner and facilitates more stable learning. It essentially reflects the MLLM's confidence in its analysis of the potential. Such a design for intermediate node rewards ensures that the tree can integrate intermediate progress into its exploration, rather than solely relying on final outcomes, thereby facilitating faster and more stable policy refinement. After calculating the node's reward, its Q-value and visit count are updated as:

$$Q_i = \frac{Q_{i-1} \times N_{i-1} + R_i}{N_{i-1} + 1}, \quad N_i = N_{i-1} + 1, \tag{2}$$

where $i$ denotes the $i$-th visit to the node, consistent with the meaning of $N_i$ (*i.e.*, $N_i = i$), $R_i$ is the reward obtained during the $i$-th visit to the node, and $Q_i$ is the node's Q-value at the $i$-th visit. Finally, the process enters the backpropagation phase. Based on the reward computed in the simulation phase, the values of the nodes are updated upwards along the search path, starting from the newly expanded node. These four phases are repeated in a cycle until an interaction trajectory that matches the target intent is successfully mined. Overall, our MCTS framework leverages three specialized agents: the InferAgent for action generation (expansion phase), the OrchestraAgent for merging and prioritizing actions (expansion phase), and the JudgeAgent for reward estimation (simulation phase). For more details of our agents implementation and MCTS phases, please refer to Appendix A.2. Additionally, visualization of the mining process is available in the Appendix C.3.

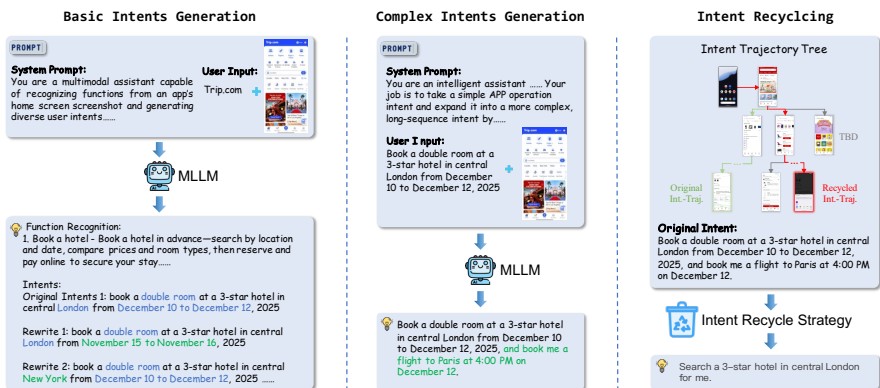

Figure 3: **Intent Generation**. Our intent generation method consists of three stages: basic intents generation, complex intents generation, and intent recycling.

### 3.3 INTENT RECYCLING STRATEGY

For a given intent, once mining is complete, a corresponding trajectory tree is formed. We discover that within this tree, besides the single trajectory pertaining to the original intent, other trajectories possess the potential to generate new intents. For instance, in a map application, while the original intent to be mined is querying routes between two locations, an incorrect tap on the "book a ride" button might lead to an additional ride-booking intent trajectory in the intent trajectory tree. These new intent-trajectory pairs do not require re-mining, which can significantly boost mining efficiency, increase intent diversity, and alleviate the difficulty of intent construction. To fully leverage these valuable trajectories retained within the tree, we propose an intent recycling strategy, as illustrated in Fig. 2(b). Specifically, for each finished tree, we consider the paths from the root to every node as the potentially valuable trajectories. To filter out low-quality trajectories, we construct an intent recycling filter using the MLLM to assess and score the quality of these trajectories. For the trajectories that passed the filter, we use the MLLM to generate intents that align with the trajectories, thereby obtaining potential intent-trajectory pairs. Subsequently, we employ the JudgeAgent to assess the status of the trajectory's last node. If it is *success*, we consider the trajectory valid which is then recycled and utilized in subsequent GUI agent training. This intent recycling strategy enables our $M^2$-Miner to fully leverage every search attempt during the MCTS process, acquiring unexpected interaction trajectory data. Details of the prompt templates are given in Appendix A.5, while examples illustrating the multi-intent trajectory tree can be found in Appendix C.2.

### 3.4 MODEL-IN-THE-LOOP TRAINING STRATEGY

During the mining process, we observe that insufficient capabilities of the InferAgent and JudgeAgent led to numerous invalid explorations and inaccurate termination judgments, which resulted in a low mining success rate. Consequently, we design a progressive model-in-the-loop training strategy that iteratively improves the agents' performance. First, in the **warm-up** stage, we first train the InferAgent and JudgeAgent using public datasets to equip the multi-agent framework with basic capability for trajectory mining. Subsequently, leveraging the models obtained in the warm-up stage, we conduct continuous training of the InferAgent and JudgeAgent through three stages. At each stage, we first generate intents, then use the mining framework to mine trajectory data, and subsequently retrain the model with all intent trajectory data mined prior to this stage. Fig. 3 illustrates the intent generation process at each stage. We will detail these three training stages below.

**Stage 1: Training on basic intents.** In this stage, we first collect the home screen screenshots of popular apps, and then, summarize the commonly used services of these applications and generate user intents based on MLLM. Finally, we perform intent expansion through conditional rewriting (*e.g.*, modifying time, location). For example, *"book a double room at a 3-star hotel in central London from December 10 to December 12, 2025."* can be rewritten as *"book a double room at a 3-star hotel in central London from November 15 to November 16, 2025."* Next, we utilize our proposed MCTS-based collaborative multi-agent framework to mine the trajectory data of these new intents. In this way, we obtain abundant intent trajectory data, which is continuously used to train the two agents to improve their performance.

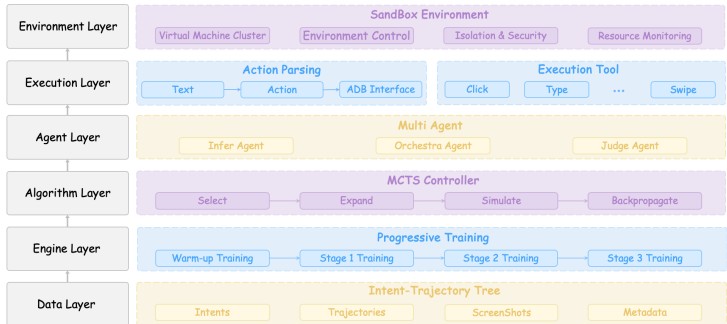

Figure 4: **Infrastructure Framework**. A layered framework for GUI agent data mining with model-in-the-loop training, comprising data, engine, algorithm, agent, execution, and environment layers.

**Stage 2: Training on complex intents.** We generate complex intents by adding conditions and functional combinations to the common user intents extracted in the previous stage. For example, the intent *"book a double room at a 3-star hotel in central London from December 10 to December 12, 2025."* can be extended to *"book a double room at a 3-star hotel in central London from December 10 to December 12, 2025, and book me a flight to Paris at 4:00 PM on December 12.";* likewise, *"play a popular song for me"* can be extended to *"play a popular song for me and share it with my friends"*. These long-sequence intents, together with the previously failed intents, form a set of complex intents that are mined using our multi-agent framework. The trajectory data obtained from this mining process are subsequently used for the training of both the InferAgent and the JudgeAgent. This process can be repeated in an alternating iteration between re-mining failed intents and continuously training the agents, thereby progressively improving the overall performance of the framework. Prompt templates for the two stages are detailed in Appendix A.6.

**Stage 3: Training on recycled intents.** We adopt the intent recycling strategy from Sec. 3.3 to review all previous intent trajectory trees, thereby enriching the trajectories. Similar to the earlier stages, both agents are continuously trained on all mined trajectory datasets. This iterative alternation between trajectory mining and model training forms a positive feedback, model-in-the-loop strategy that progressively improves mining success rates and data quality. The resulting multi-agent framework effectively mines trajectories.

## 4 EXPERIMENTS

### 4.1 INFRASTRUCTURE FRAMEWORK

We build a customized infrastructure framework that supports mobile agent data mining and model-in-the-loop training. As illustrated in Fig. 4, the framework adopts a layered architecture, consisting of the data layer, engine layer, algorithm layer, agent layer, execution layer, and environment layer. Our layered framework unifies GUI agent execution, intent-trajectory data mining, and model training into a whole, thereby forming a customized model-in-the-loop framework that can automatically perform end-to-end data mining and training. More details of the infrastructure and environment interaction are provided in Appendix B.7.

### 4.2 $M^2$-MINER-AGENT DATASET STATISTICS

By employing our automated data mining framework together with minimal manual quality inspection, we collect the **$M^2$-Miner-Agent** dataset (abbreviated as **$M^2$**), which consists of 20k images and 2,565 trajectories, with an average trajectory length of 7.8. Compared to open-source datasets such as Android Control Li et al. (2024), AITZ Zhang et al. (2024), AMEX Chai et al. (2025), and GUI Odyssey Lu et al. (2025a), which rely on labor-intensive manual annotation and quality review processes, our dataset exhibits a significantly lower collection cost. This is crucial when targeting new mobile applications. To fairly evaluate the cost of dataset construction, we estimate the costs for all datasets based on the hourly wage rate ($r_{\text{wage}} = \$7/h$), annotation efficiency ($t_{\text{annot}} = 0.05 h/image$), and quality inspection efficiency ($t_{\text{inspect}} = 0.0014 h/image$), provided by a professional annotation team. Table 1 summarizes the key statistics of $M^2$-Miner-Agent and other datasets. In comparison to AITZ, which is of a similar scale, our method reduces construction

Table 1: Statistics of different datasets. **Size**: number of images; **Su.RL**: suitability for reinforcement learning algorithms; **Auto**: auto-annotated data; **AvgStp**: average step length; **Trajs**: number of trajectories; **Cost**: estimated data production cost (USD). **Cost per image**: estimated cost of producing one image (USD).

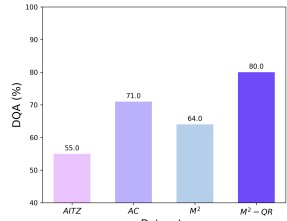

Figure 5: Evaluation of data quality. $M^2$: our mined dataset, $M^2$-**QR**: our dataset after human quality review.

| Dataset | Size | Su.RL | Auto | AvgStp | Trajs | Cost | Cost per image |
|---|---|---|---|---|---|---|---|
| Android Control | 88k | ✗ | ✗ | 5.5 | 15,283 | 31,662 | 0.36 |
| AMEX | 38k | ✗ | ✗ | 12.8 | 2,946 | 13,680 | 0.36 |
| GUI-Odyssey | 119k | ✗ | ✗ | 15.4 | 7,735 | 42,816 | 0.36 |
| AITZ | 18k | ✗ | ✗ | 7.5 | 2,504 | 6,476 | 0.36 |
| **$M^2$-Miner-Agent** | 20k | ✓ | ✓ | 7.8 | 2,565 | 466 | 0.02 |

costs by \$6,010. When comparing on per-image cost (calculated as total cost divided by the number of images), our method is approximately 18 times more cost-effective than all other datasets. The cost calculation formula and a detailed explanation are provided in Appendix B.6. To evaluate the quality of our mined data, we adopt a sample-and-verify approach. Specifically, we randomly select 100 sequence samples from each of the AC, AITZ, and our dataset, and conduct manual inspection to verify their correctness. As shown in Fig. 5, the Data Quality Accuracy (DQA) of our mined and inspected data is higher than that of manually annotated datasets. Examples of incorrectly annotated samples from public data are provided in Appendix C.1.

## 4.3 EXPERIMENT SETUP

**Implementation Details.** We use Qwen2.5-VL-7B Bai et al. (2025) as the base model for both the InferAgent and JudgeAgent. The OrchestraAgent, the intent recycling filter, and the MLLM for generating new intents based on recycled trajectories are all implemented with Qwen2.5-VL-72B. More training details and prompt templates are provided in Appendix B.1 and Appendix A.2.

**Evaluation Benchmarks.** We evaluate our method on four representative GUI interaction benchmarks: AC Li et al. (2024), AITZ Zhang et al. (2024), GUI Odyssey Lu et al. (2025a) and CAGUI Zhang et al. (2025b). Since the training set of CAGUI has not been released, it can be used to evaluate the generalization ability of our mining framework in novel scenarios. For detailed information regarding the benchmarks and evaluation protocols, please refer to the Appendix B.3.

**Metrics.** We assess the success rate and quality of mined data using two measures: Mining Success Ratio (MSR), which quantifies the proportion of successful mining attempts, and Data Quality Accuracy (DQA), which evaluates the correctness of the collected data. For the evaluation of GUI agents, we further utilize two mainstream metrics for GUI agent evaluation: the accuracy of action type prediction (TP) and the step success rate (SR). Appendix B.4 details the metrics.

**Baselines.** We compare our method with a wide range of baselines, including various proprietary MLLMs, open-source GUI agent models, and representative models trained on automatically mined data, such as OS-Genesis-7B Sun et al. (2025) and GUI-Owl-7B Ye et al. (2025). Baseline results and evaluation protocols follow the original paper to ensure fairness. More details about the baselines can be found in Appendix B.5.

## 4.4 MAIN RESULTS

To validate the effectiveness of our method, we use public datasets and mined data to train Qwen2.5-VL-3B and Qwen2.5-VL-7B, obtaining $M^2$-**Miner-3B** and $M^2$-**Miner-7B**. We evaluate against prior SOTA methods on the GUI agent task. Quantitative results in Table 2 show that our approach outperforms baselines across all benchmarks and achieves a new SOTA. For AC-Low, $M^2$-Miner-7B achieves 97.5% on TP and 93.5% on SR, surpassing all previous methods by a large margin. While ranking second in TP (81.8%) on AC-High, we achieve the best SR (72.9%), indicating superior task completion. On the challenging AITZ benchmark, our model attains 81.3% on TP and 69.4% on SR, achieving a new SOTA result. For CAGUI, Qwen2.5-VL-7B trained with our mined data boosts SR from 55.2% to 70.2%, highlighting strong generalization to unseen domains. Moreover, our model consistently outperforms OS-Genesis-7B Sun et al. (2025) and GUI-Owl-7B Ye et al. (2025), the primary baselines employing automatically mined data. While UI-TARS-7B Qin et al. (2025), which is trained on a large-scale private human-annotated dataset, exhibits strong performance, our

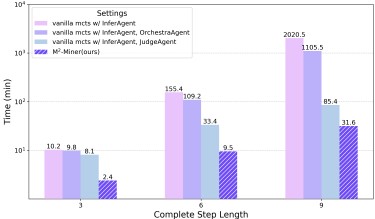 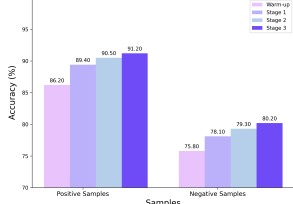 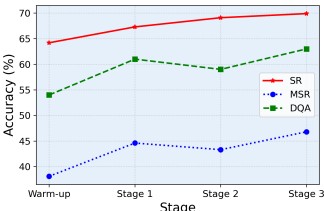

Figure 6: Ablation study of the collaborative multi-agent framework for accelerating MCTS-based data mining. For better visualization, a logarithmic scale was used.

Figure 7: Ablation study on model-in-the-loop for JudgeAgent. The performance of positive and negative samples is reported.

Figure 8: Ablation study on model-in-the-loop for data quality. The performances of SR, MSR and DQA at different stages are reported.

Table 2: Performance comparison on GUI agent benchmarks (AC-Low, AC-High, AITZ, GUI-Odyssey, and CAGUI). **Bold** and underline indicate the best and second-best results.

| Models | AC-Low | | AC-High | | AITZ | | GUI-Odyssey | | CAGUI | |
|---|---|---|---|---|---|---|---|---|---|---|
| | TP | SR | TP | SR | TP | SR | TP | SR | TP | SR |
| *Closed-source Models* | | | | | | | | | | |
| GPT-4o Hurst et al. (2024) | 74.3 | 19.4 | 66.3 | 20.8 | 70.0 | 35.3 | - | - | 3.67 | 3.67 |
| Gemini 2.0 Deepmind (2024) | - | 28.5 | - | 60.2 | - | - | - | - | - | - |
| Claude Anthropic (2024) | 74.3 | 19.4 | 63.7 | 12.5 | - | - | 60.9 | - | - | - |
| *Open-source Models W/ Private Human-Annotated Datasets* | | | | | | | | | | |
| UI-TARS-7B* Qin et al. (2025) | 98.0 | 90.8 | 83.7 | 72.5 | 80.4 | 65.8 | 90.1 | 87.0 | 88.6 | 70.0 |
| *Open-source Models W/ Public Datasets* | | | | | | | | | | |
| OdysseyAgent Lu et al. (2025a) | 65.1 | 39.2 | 58.8 | 32.7 | 59.2 | 31.6 | - | 78.2 | 67.6 | 25.4 |
| SpiritSight-8B Huang et al. (2025) | - | 87.6 | - | 68.1 | - | - | - | 75.8 | - | - |
| GUI-R1-7B Luo et al. (2025) | 85.2 | 66.5 | 51.7 | 56.8 | 50.5 | 65.5 | 38.8 | - | - | |
| OS-Atlas-7B Wu et al. (2025a) | 93.6 | 85.2 | **85.2** | 71.2 | 74.1 | 58.5 | 84.5 | 62.0 | 81.5 | 55.9 |
| Aguvis-7B Xu et al. (2025b) | - | 80.5 | - | 61.5 | 35.7 | 19.0 | 26.7 | 13.5 | 67.4 | 38.2 |
| InfiGUI-R1 Liu et al. (2025) | 96.0 | 92.1 | 82.7 | 71.1 | 70.7 | 52.9 | 74.5 | 55.0 | - | - |
| Qwen2.5-VL-3B Bai et al. (2025) | 92.2 | 80.4 | 76.5 | 60.2 | 75.1 | 52.7 | 81.2 | 57.8 | 71.9 | 53.1 |
| Qwen2.5-VL-7B Bai et al. (2025) | 94.1 | 85.0 | 75.1 | 62.9 | 78.4 | 54.6 | 83.7 | 60.3 | 74.2 | 55.2 |
| *Open-source Models W/ Auto-Mined Datasets* | | | | | | | | | | |
| OS-Genesis-7B Sun et al. (2025) | 90.7 | 74.2 | 66.2 | 44.5 | 20.0 | 8.5 | 11.7 | 3.6 | 38.1 | 14.5 |
| GUI-Owl-7B Ye et al. (2025) | 93.8 | 90.0 | 81.5 | 72.8 | 78.9 | 65.1 | 83.4 | 60.7 | 80.0 | 59.2 |
| **M²-Miner-3B** | 97.2 | 93.2 | 81.3 | 71.2 | 78.6 | 66.6 | 88.2 | 77.1 | 88.5 | 67.3 |
| **M²-Miner-7B** | **97.5** | **93.5** | 81.8 | **72.9** | **81.3** | **69.4** | **90.5** | **79.3** | **88.8** | **70.2** |

model not only achieves comparable results on TP but also consistently surpasses UI-TARS-7B in SR across almost all benchmarks. Appendix C.4 presents qualitative results of M²-Miner-7B.

## 4.5 ABLATION STUDIES

**Ablation on Collaborative Multi-Agent Framework.** We perform an ablation study on the multi-agent framework (Sec. 3.2) to assess the contribution of each agent to mining efficiency. For vanilla MCTS with only the InferAgent, the InferAgent will generate several actions during the expansion phase, but these actions are often redundant (e.g., clicking the same button at different coordinates) and are not ranked by priority. The simulation phase is the same as vanilla MCTS, where node values are evaluated via costly rollouts. Rollouts are executed by continuously executing the InferAgent until it outputs success, failure, or reaches the preset maximum number of steps. When the OrchestraAgent is introduced in the expansion phase, it merges the redundant actions generated by the InferAgent and prioritizes correct ones, thereby reducing exploration nodes. When the JudgeAgent is introduced in the simulation phase, rollouts are no longer needed. Instead, the node values are directly evaluated by the JudgeAgent, significantly reducing simulation time. As shown in Fig. 6, compared with vanilla MCTS using only the InferAgent, the efficiency improvements brought by M²-Miner grows exponentially with task complexity, achieving a $64\times$ speedup at task length 9. Furthermore, the results show that each agent is essential for boosting mining efficiency.

**Ablation on Model-in-the-loop.** To demonstrate the effectiveness of our proposed progressive model-in-the-loop training strategy (Sec. 3.4), we conduct an ablation study on the CAGUI bench-

---
*UI-TARS-7B uses a large amount of private human-annotated data and is not included in this comparison to ensure fairness.

Table 3: Ablation on model-in-the-loop. **TBI**: Training on basic intents; **TCI**: Training on complex intents; **TRI**: Training on recycled intents.

| Stage | TBI | TCI | TRI | TP | SR |
|---|---|---|---|---|---|
| Warm-up | | | | 85.0 | 64.2 |
| Stage 1 | ✓ | | | 86.5 | 67.3 |
| Stage 2 | ✓ | ✓ | | 87.6 | 69.1 |
| Stage 3 | ✓ | ✓ | ✓ | 88.2 | 69.9 |

Table 4: Ablation on data structure. **ACT**: data containing only actions; **DES**: data containing descriptions; **PREF**: preference data.

| Setting | ACT | DES | PREF | TP | SR |
|---|---|---|---|---|---|
| Act. | ✓ | | | 85.2 | 66.8 |
| Act. + Des. | ✓ | ✓ | | 88.2 | 69.9 |
| Act. + Des. + Pref. | ✓ | ✓ | ✓ | 88.8 | 70.2 |

Table 5: Ablation on training datasets. The results show overall TP, SR, and per-action accuracy.

| Model | Training Dataset | TP | SR | Click | Scroll | Type | Press | Stop |
|---|---|---|---|---|---|---|---|---|
| Qwen2.5-VL-7B | – | 74.2 | 55.2 | 55.6 | 11.3 | 41.9 | 0.0 | 61.1 |
| Qwen2.5-VL-7B | Public datasets | 84.9 | 64.4 | 67.1 | 22.3 | 62.3 | 0.0 | 65.5 |
| Qwen2.5-VL-7B | Auto-Mined Datasets | 87.1 | 69.5 | 70.7 | 24.7 | 68.5 | 0.0 | 78.5 |
| Qwen2.5-VL-7B | Public and Auto-Mined Datasets | 88.8 | 70.2 | 71.2 | 26.6 | 69.5 | 0.0 | 72.8 |

mark, which has no training data and is therefore well-suited for simulating data mining in new scenarios. As shown in Table 3, benefiting from the newly mined trajectory data at each stage, the model's performance is naturally enhanced, which in turn facilitates the mining process. Fig. 7 indicates that JudgeAgent's performance in distinguishing positive and negative samples exhibits a steady upward trend. Furthermore, Fig. 8 shows that, although the data mining success rate at the warm-up stage is relatively low, our approach yields a remarkable improvement in both MSR and DQA in all subsequent stages compared with the warm-up stage. Stage 2 focuses on mining complex intents, which results in only a modest improvement in MSR. Nevertheless, the data generated in this stage also significantly enhanced the TP and SR of downstream tasks. In addition, in the stage 3, when recycling diverse intents, both MSR and DQA improve significantly. These results indicate that our progressive model-in-the-loop strategy has significant advantages in improving TP, SR, MSR, and DQA. The data construction method for JudgeAgent is provided in Appendix A.4.

**Ablation on Data Structure.** Manually annotated datasets typically contain only simple action labels and lack semantic information. In contrast, the trajectory trees preserved during our mining process contain both actions and descriptions with rich semantic content. Moreover, by leveraging the positive samples from the trajectory trees and the negative samples from other branches, preference data can be constructed. The results in Table 4 demonstrate that the description data and preference data obtained during our mining process can effectively improve the TP and SR.

**Ablation on Training Datasets.** We conduct an ablation study on the training data, and the results are presented in Table 5. The experiments are performed on the CAGUI benchmark using the Qwen2.5-VL-7B model. Without training data, the Qwen2.5-VL-7B model shows relatively low accuracy across all action types (CLICK, SCROLL, TYPE and STOP), indicating limited execution performance on new GUI applications. Training on public datasets produces moderate improvements, with SR increasing by 9.2% and performance gains observed across all action types. Using only auto-mined data results in larger boosts (SR +14.3%, CLICK +15.1%, SCROLL +13.4%, TYPE +26.6% and STOP +17.4%), highlighting the practical significance of our proposed data production framework. When combining public and auto-mined datasets, the model achieves the best results, outperforming the public-only setting by +3.9% TP, +5.8% SR, and showing consistent gains across all actions. In summary, the ablation results show that our auto-mined data serves as a valuable complement to existing public datasets and highlight our core contribution: an automated data mining framework, which has practical significance for the GUI Agent domain.

## 5 CONCLUSION

In this work, we propose $M^2$-Miner, a fully automated framework for mobile GUI agent data mining. By introducing MCTS and designing a collaborative multi-agent framework, our method significantly improves data mining efficiency while enhancing data quality. The intent recycling strategy further enhances both mining efficiency and intent richness, while the progressive model-in-the-loop training paradigm boosts success rates in both familiar and novel environments. Extensive experiments show that GUI agents trained on our mined data achieve SOTA performance. $M^2$-Miner provides a promising approach for mining high-quality trajectory data for mobile GUI agent, establishing a solid foundation for the development of GUI research community.

ACKNOWLEDGMENTS

This work was supported by Ant Group and in part by Zhejiang Province Program (2024C03263).

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

## STATEMENT OF USING LLM

We use LLM to optimize our English expressions for improved clarity.

## APPENDIX OUTLINE

Here we provide implementation details (Appendix A), experimental details (Appendix B) and visualization results (Appendix C) omitted from the main paper for brevity.

Specifically, Appendix A.1 introduces the preliminaries for the canonical Monte Carlo Tree Search algorithm. Appendix A.2, A.3, and A.4 describe the details of InferAgent, OrchestraAgent, and JudgeAgent, respectively. Appendix A.5 details our intent recycling strategy. Appendix A.6 introduces model-in-the-loop details.

Appendix B.1 and B.2 present the training details and the action space, respectively. Appendix B.3, B.4 and B.5 introduce benchmark datasets, evaluation metrics and baselines. Appendix B.6 describes calculation of dataset construction cost. Appendix B.7 details infrastructure framework and environment interaction process.

Appendix C.1 shows erroneous samples in public datasets. Appendix C.2 presents examples of intent-trajectory trees. Appendix C.3 shows examples of data mining process. Appendix C.4 presents GUI agent inference examples on benchmark datasets. Besides, we have provided a supplementary video that demonstrates the data mining process.

## A   IMPLEMENTATION DETAILS

### A.1   PRELIMINARY: MONTE CARLO TREE SEARCH

Monte Carlo Tree Search is a heuristic tree search algorithm widely used in sequential decision-making tasks. Unlike traditional tree search methods that rely on handcrafted evaluation functions, MCTS dynamically constructs a search tree through iterative simulations, balancing exploration and exploitation using statistical estimates. Each node in the tree represents a state in the environment, and each edge corresponds to an action that transitions the environment to a new state.

At the core of MCTS lies a four-step procedure repeated at each iteration: *Selection*, *Expansion*, *Simulation*, and *Backpropagation*. Specifically, during the selection phase, the algorithm traverses the search tree and selects a node using a principled selection policy. A commonly used policy is the Upper Confidence Bounds applied to Trees (UCT) Kocsis & Szepesvári (2006), which balances exploitation and exploration by choosing actions that maximize:

$$a^* = \arg \max_{a \in \mathcal{A}(s)} \left\{ \bar{Q}(s,a) + c\sqrt{\frac{\ln N(s)}{N(s,a)}} \right\}, \tag{3}$$

where $\bar{Q}(s,a)$ is the cumulative reward of the child node that is obtained by taking action $a$ in state $s$, $N(s)$ represents the visit count for state $s$, $N(s,a)$ denotes the number of times action $a$ has been taken in state $s$, and $c > 0$ is a constant controlling the exploration-exploitation trade-off. After a node is selected, an action is randomly chosen in the expansion phase to obtain a child node. Subsequently, in the simulation phase, the algorithm conducts a complete process (roll-out) starting from this newly expanded node until a terminal state is reached. This process typically follows a simple and fast policy, such as making decisions completely at random. The purpose of the roll-out is to quickly obtain a result or evaluation, such as a reward. The backpropagation phase begins after the rollout concludes and a reward is obtained. During this phase, the algorithm backtracks from the newly expanded node up to the root, utilizing the simulation results to update the statistical information of all nodes along the path, such as their visit counts and cumulative rewards (*e.g.*, number of successes).

Despite its effectiveness, vanilla MCTS suffers from computational inefficiency in large or complex environments (*e.g.*, GUI agent task). Specifically, because vanilla MCTS randomly selects actions for expansion and relies on repeated rollouts to obtain rewards, its applicability is restricted in high-dimensional or real-time applications.

## A.2  INFERAGENT

In the expansion phase of Monte Carlo Tree Search (MCTS), the InferAgent plays a pivotal role by intelligently inferring and generating the most plausible action set to achieve the target intent for each tree node. In the vanilla MCTS, random action expansion often results in inefficiency and redundant coverage of the action space. To address this, we design a multi-model collaborative mechanism for InferAgent, improving the effectiveness and diversity of expansion outcomes. Specifically, the InferAgent is composed of two models: InferAgent-Qwen-7B, which is trained on Qwen2.5-VL-7B Bai et al. (2025) using the prompt shown in Fig. 9, and InferAgent-Qwen-72B derived by instructing Qwen2.5-VL-72B Bai et al. (2025) with the prompt in Fig. 9. During the expansion of each node, we first employ InferAgent-Qwen-7B to generate one action, ensuring a reasonable exploration space. To mitigate the bias issue of the trained model and enhance the diversity of the action space, we then utilize InferAgent-Qwen-72B to subsequently generate $K-1$ actions. Notably, the prompt for InferAgent-Qwen-72B explicitly includes the list of previously generated actions to prevent the model from producing repeated actions. By introducing historical actions as constraints, this approach effectively guides the MLLM to explore novel, non-repetitive candidate actions. Furthermore, we leverage InferAgent-Qwen for reasoning in GUI agent tasks, using the same prompt template as in Fig. 9.

## A.3  ORCHESTRAAGENT

OrchestraAgent is equipped with an action merging function (derived from the prompt in Fig. 10) and an action ranking function (derived from the prompt in Fig. 11), both implemented based on Qwen2.5-VL-72B. With these two functions, OrchestraAgent can efficiently merge equivalent actions and sort candidate actions by intent relevance during the MCTS expansion phase, thereby improving exploration efficiency.

## A.4  JUDGEAGENT

JudgeAgent consists of two main modules: the Outcome Reward Model (derived from Qwen2.5-VL-72B with the prompt of Fig. 12) and the Process Reward Model (trained on Qwen2.5-VL-7B Bai et al. (2025) using the prompt of Fig. 13), which are responsible for determining the task completion status of nodes and calculating the rewards, respectively. During the simulation phase, the Outcome Reward Model is first applied to determine whether the node corresponds to a terminal state. If the status is *success* or *failure* (*i.e.* impossible_to_succeed), rewards of 1 or 0 will be assigned respectively. If the status is *intermediate* (*i.e.*, not_yet_succeeded), the Process Reward Model is further utilized to assess the potential validity (*i.e.*, valid or invalid) of this node. The logit corresponding to the token "valid" is then used as the reward signal, as described in Sec. 4.2 of the main paper. The JudgeAgent constructs positive samples based on the mined data, and identifies incorrect steps from the intent trajectory tree to serve as negative samples. The Action Type distribution of negative samples is synchronized to closely match that of the positive ones. The dataset is then divided into training and test sets in a 9:1 ratio for model training and evaluation. The distribution of the data is illustrated in Fig. 14.

## A.5  INTENT RECYCLING STRATEGY

As detailed in Sec. 4.3, for each completed trajectory tree, we enumerate all root-to-node paths as candidate trajectories. Subsequently, we employ the recycling filter (derived from Qwen2.5-VL-72B with the prompt of Fig. 15) to evaluate the quality and relevance of each candidate trajectory, considering the UI context and action sequences, and filter out low-quality trajectories. For the trajectories that pass this filter, we further instruct Qwen2.5-VL-72B with the prompt of Fig. 16 to generate corresponding intent descriptions. Finally, JudgeAgent evaluates whether the reviewed trajectory fulfills its associated intent, retaining only those samples deemed successful. By applying such strategy, $M^2$-Miner fully leverages every search process, significantly improving mining efficiency and enriching the diversity of intents.

Figure 9: Prompt for InferAgent-Qwen.

## A.6 MODEL-IN-THE-LOOP

As illustrated in Fig. 17 and Fig. 18, we constructed two types of prompts: one for basic intent generation and another for complex intent generation. Leveraging the Qwen2.5-VL-72B model, we produced intent datasets in two successive stages corresponding to these prompt types.

## B EXPERIMENTAL DETAILS

### B.1 TRAINING DETAILS

The InferAgent and JudgeAgent are trained iteratively with our model-in-the-loop training strategy. Specifically, in the warm-up stage, training data are drawn from public datasets, including Android-

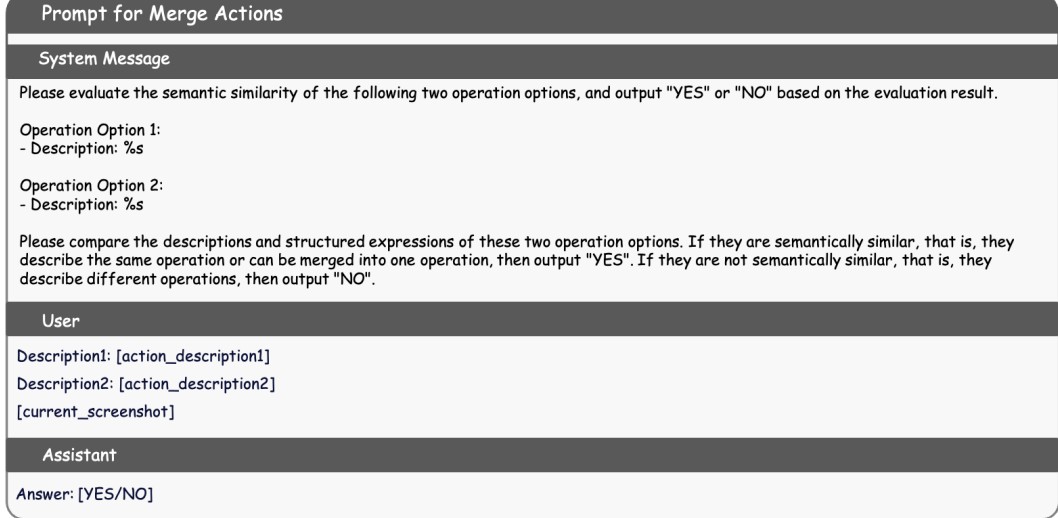

Figure 10: Prompt for Action Merging Function of OrchestraAgent.

Table 6: Action Space for Mobile GUI Agent Tasks. **Coord.**: coordinate

| Action Type | Description | Required Parameters |
|---|---|---|
| click | Tap at a specific Coord. $(x, y)$ | Coord.: $[x, y]$ |
| long_press | Long press at a point for a specified duration | Coord.: $[x, y]$, time: seconds |
| swipe | Swipe from one point to another | Coord.: $[x_1, y_1]$, Coord.2: $[x_2, y_2]$ |
| type | Type text into the current input field | text: string |
| key | Simulate key events (*e.g.*, volume_up, power) | text: key name |
| wait | Wait for a specified number of seconds | time: seconds |
| system_button | Press a system button (Back, Home, etc.) | button: Back, Home, Menu, or Enter |
| terminate | Terminate the current task and report completion status | status: success or failure |

Control Li et al. (2024), AITZ Zhang et al. (2024), GUI-Odyssey Lu et al. (2025a), and AMEX Chai et al. (2025). All training is conducted on 8 NVIDIA A100-80G GPUs; at each stage we retrain for 2 epochs on the full mined dataset, using a global batch size of 8 and a learning rate of $3 \times 10^{-5}$. We apply LoRA fine-tuning to all linear modules with a rank of 32 and $\alpha$ of 64.

## B.2 ACTION SPACE

Table 6 lists all the action spaces used by M²-Miner. Specifically, these action types comprehensively encompass common interactions on mobile devices, such as tapping or long pressing at specific coordinates, swiping between points, typing text, simulating key events, pressing system buttons, waiting, and terminating tasks. Each action requires clearly defined parameters, ensuring precise and reproducible interaction modeling.

## B.3 BENCHMARKS

Table 7 presents detailed statistics on the composition of the datasets used. The division between the training and test sets strictly follows the official split to ensure consistency with the original data. We provide more information about the utilized datasets below. **AndroidControl** (**AC**) Li et al. (2024) is a comprehensive and widely adopted benchmark for assessing real-world mobile GUI navigation agents. The dataset is constructed from human-collected interaction tasks within the Android environment, encompassing 15,283 trajectories from 833 different apps. Each episode contains, on average, 5.5 fine-grained annotated action steps, enabling detailed evaluation at both the action type and execution levels for single-app scenarios. For evaluation, AndroidControl sup-

**Prompt for Rank Actions**

**System Message**

You are an expert in mobile phone automation who is functioning as a critic. You will be shown a few possible tasks that can be done on a mobile phone in order to move towards an obejctive and you have to review which of them is the best suited one to achieve the said objective. You will be given the main objective, the screenshot of the mobile phone on which these tasks are supposed to be executed and the past history of execution.
### Execution Flow Guidelines
- You will have a look at the objective that needs to be achieved.
- Then, you will look at the tasks that have been done till now, their successes/ failures. If no tasks have been completed till now, that means the system has started from scratch.
- Post this, have a careful look at the current screenshot provided to you. Use that to see if the actions in the proposed tasks have the correct elements that they are supposed to act on.
- Once you have carefully observed the screenshot, previous tasks and the objective, think step by step and choose the best possible next step out of the given possible tasks that can be executed on the current screenshot in order to move towards the overall objective. Think of these given tasks like branches from the same root node(the webpage) - like three different paths that eventaully should lead to the overall objective. You should act like a critic and carefully follow the below instructions.
Your input and output will strictly be a well-formatted JSON with attributes as mentioned below.
### Input Format
- objective: Mandatory string representing the main objective to be achieved via mobile phone automation
- tasks_before: Optional summary (string format) list of completed tasks contributing to objective fulfillment.
- tasks_for_eval: Mandaory List of possible next tasks of which anyone can be done on the screenshot to achieve/ move towards the objective. Think step by step. Choose one of these based on the overall objective, tasks completed till now and the current state of the mobile phone. You will be provided with a screenshot of the mobile phone to think better.
#### Format of Task Object in tasks_for_eval:
- id: Mandatory Integer representing the id of the task
- description: Mandatory string representing the description of the task
- action: A JSON object indicating the action that need to be execute in order to complete the task with its appropriate fields.
### Output Format
- thought - A Mandatory string specifying your thoughts on how did you come up with top task. reiterate the objective here so that you can always remember what's the system's eventual aim. Act like a critic, reason deeply about the possible flaws in each option and think step by step to come up with one top task. Illustrate your thoughts here.
- top_task_id: id of the task you think is the best suited one to be performed on the screenshot to lead towards/ complete the objective

**User**

Objective: [objective_info]
tasks_before: [tasks_before_list]
tasks_for_eval: [tasks_for_eval_list]
[current_screenshot]

**Assistant**

Thought: [thought]
Top_task_id: [task_id]

Figure 11: Prompt for Action Ranking Function of OrchestraAgent.

Table 7: Overview of GUI agent datasets. **AvgStp** denotes the average step length. **Total Traj.** represents the total number of trajectories. **Train Split** and **Test Split** specify how many trajectories are allocated for model training and evaluation respectively.

| Dataset | Platform | AvgStp | Total Traj. | Train Split | Test Split |
|---|---|---|---|---|---|
| AndroidControl Li et al. (2024) | Mobile | 5.5 | 15,283 | 13,602 | 1,681 |
| AITZ Zhang et al. (2024) | Mobile | 6.0 | 2,504 | 1,998 | 506 |
| GUI Odyssey Lu et al. (2025a) | Mobile | 15.3 | 7,735 | 5,801 | 1,934 |
| AMEX Chai et al. (2025) | Mobile | 12.8 | 2,946 | 2,946 | 0 |
| CAGUI Zhang et al. (2025b) | Mobile | 7.5 | 600 | 0 | 600 |

ports two settings: AndroidControl-High (AC-High), where each prompt presents only a high-level instruction, and AndroidControl-Low (AC-Low), which provides both high-level and corresponding low-level instructions. The benchmark adopts success rate (SR) and action type accuracy (Type)

---

**Prompt for Outcome Reward Model**

**System Message**

You are an expert mobile automation judge. Your task is to evaluate whether an AI agent has achieved the user's objective, using the user's objective description, the current screenshot, and the agent's completed actions.

## Evaluation States
1. **Success** - Thoroughly verify that all key elements explicitly stated in the user's objective (such as time, location, origin, destination, subject, etc.) are present. Only judge it as Success if every requirement is fully satisfied. - The screenshot must clearly display the exact page or state requested, including all specified details. - The AI must have accurately navigated to the intended target state. - Example: If the objective is "Go to payment page for flight from Beijing to Shanghai on June 12," the screenshot must show the payment page with June 12 as the date, Beijing as the origin, and Shanghai as the destination. - Output: `{"thought": "your reasoning", "is_terminal": true, "status": "success"}`
2. **Not Yet Succeeded** - The agent has not yet reached the final target page or state, but it is still possible to continue and achieve the objective. - If any key condition specified in the objective is not yet satisfied, select this state. - Output: `{"thought": "your reasoning", "is_terminal": false, "status": "not_yet_succeeded"}`
3. **Impossible to Succeed** - The agent has reached an error or a dead end where it is impossible to achieve the objective. - This includes cases where continuing would require user credentials, payments, or private information that the AI cannot provide. - Output: `{"thought": "your reasoning", "is_terminal": false, "status": "impossible_to_succeed"}`

## Evaluation Process
1. Identify the key required elements in the objective (such as time, location, origin, destination, subject, etc.).
2. Compare the screenshot and completed actions to check if all required elements have been fulfilled.
3. If all conditions are met, output Success. If requirements are not yet met but achieving the objective is still possible, output Not Yet Succeeded. If the objective is impossible to achieve, output Impossible to Succeed.
4. In all cases, explain your reasoning in the "thought" field before giving your final decision.

## Output Format
Return a JSON object with exactly three keys: - `"thought"`: A string explaining the reasoning behind your judgment - `"is_terminal"`: Boolean indicating if the process has reached a final state (success or impossible to continue) - `"status"`: String, one of: `"success"`, `"not_yet_succeeded"`, or `"impossible_to_succeed"`

**User**

The user query : [user_request]
Actions: [all_history_actions]
[current_screenshot]

**Assistant**

Thought: [thought]
Is_terminal: [true/false]
Status: [success/not_yet_succeeded/impossible_to_succeed]

Figure 12: Prompt for Outcome Reward Model of JudgeAgent.

as its main metrics, leveraging ground truth action labels for quantitative assessment. The official train/test split allocates 13,602 trajectories for training and 1,681 for testing.

**AITZ** Zhang et al. (2024) is a task-oriented mobile interaction dataset comprising 2,504 unique instructions and 18,643 screen–action pairs, along with four types of semantic annotations, covering more than 70 Android applications. Each operation trajectory contains an average of 7.5 steps, reflecting real-world user behaviors and dynamic UI scenarios, and supporting robust contextual reasoning for agent evaluation. In our experiments, we follow the official split with 1,998 trajectories in the training set and 506 in the test set.

**GUI Odyssey** Lu et al. (2025a) is a large-scale benchmark targeting complex cross-app mobile tasks. It contains 8,334 episodes (7,735 trajectories after semantic processing Xu et al. (2025b)), with an average of 15.3 steps per episode, covering 6 mobile devices, 212 distinct apps, and 1,357 app combinations. Each step is enriched with detailed semantic reasoning annotations, which help agents form cognitive processes and strengthen reasoning for multi-step and cross-application navigation. We follow the official in-domain split protocol, resulting in 5,801 trajectories for training and 1,934

**Prompt for Process Reward Model**

**System Message**

You are an action validator for GUI automation systems. Your role is to evaluate whether a proposed action is a logical and appropriate step toward completing the given human instruction.

## Core Responsibilities:
- Validate that the current action makes progress toward the stated goal; Consider the context provided by past actions; Evaluate feasibility and appropriateness of the action; Determine if the action follows logical UI interaction patterns.

Validation Criteria:
**VALID actions must:**
- Directly contribute to completing the human instruction; Be contextually appropriate given previous actions; Follow standard GUI interaction patterns; Target the correct UI elements for the intended task; Represent reasonable next steps in the workflow.
**INVALID actions include:**
- Actions that contradict the human instruction; Redundant actions that repeat previous steps unnecessarily; Actions targeting incorrect or non-existent UI elements; Steps that would break the current workflow; Actions that move away from the stated objective.

## Context Usage:
- Past actions provide workflow context but don't need to be repeated; Focus on whether the current action logically follows from the instruction and context; Consider the current state of the application/interface

## Action Space:
The following actions are available in the mobile GUI automation system:
**Available Actions:**
- `key`: Perform a key event (supports adb keyevent syntax: "volume_up", "volume_down","power","camera","clear"); `click`: Click at coordinate (x, y) on screen; `long_press`: Press at coordinate (x, y) for specified seconds; `swipe`: Swipe from coordinate (x, y) to coordinate2 (x2, y2); `type`: Input text into activated input box; `system_button`: Press system button ("Back", "Home", "Menu", "Enter"); `open`: Open an app on the device; `wait`: Wait specified seconds for changes; `terminate`: End task with status ("success" or "failure")

## Output Requirements:
- Respond with exactly one word: "valid" or "invalid"
- No explanations, reasoning, or additional text
- Base decision on logical progression toward the human instruction

**User**

The user query : [user_request]

History actions: [history_actions]

Current action to validate: [current_action]

[current_screenshot]

**Assistant**

Content: [valid/invalid]

Figure 13: Prompt for Process Reward Model of JudgeAgent.

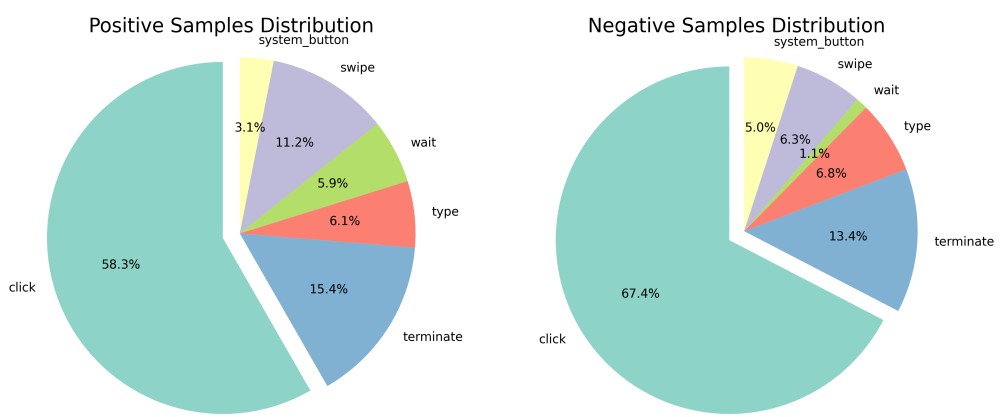

Figure 14: Distribution diagram of positive and negative sample data for the JudgeAgent.

for testing. **AMEX** Chai et al. (2025) is a mobile GUI dataset focused on evaluating agent execution under constrained or domain-specific contexts. It contains 2,991 trajectories averaging 11.9 steps each, all allocated to training, with no official test set provided.

**Prompt for Intent Recycling Filter**

**System Message**

# # GUI Agent Path Quality Assessment Expert

You are a professional GUI automation path quality assessment expert. Your task is to analyze GUI operation execution paths provided by users and evaluate whether there are redundant or erroneous steps in the paths.

## Assessment Target Description

- **Path Structure**: Linear operation sequences where each node contains:
  - Action Description: GUI operations from previous node to current node (click, swipe, input, back, etc.)
  - GUI Final State: Screenshot state after executing all operations

## Core Assessment Dimensions

### 1. Operation Redundancy Analysis

- **Duplicate Operations**: Whether there are consecutive or intermittent identical operations
- **Invalid Loops**: Whether there are meaningless operations like entering an interface and immediately exiting
- **Unnecessary Steps**: Whether there are operation sequences that could achieve the same effect through shorter paths
- **State Rollbacks**: Whether there are unnecessary operations of returning to previous level and re-entering

### 2. Operation Correctness Verification

- **Context Adaptation**: Whether operations match the current GUI state
- **Element Validity**: Whether there are operations clicking non-existent or non-interactive elements
- **Input Rationality**: Whether text input and selection operations comply with interface requirements
- **State Consistency**: Whether the GUI state changes are consistent with the performed operations

### 3. Logical Flow Reasonableness

- **Operation Sequence**: Whether the step order follows normal user behavior logic
- **State Transitions**: Whether interface state changes after each operation are reasonable
- **Dependency Relations**: Whether prerequisite relationships between operations are followed
- **Exception Handling**: Whether unnecessary error recovery operations are included

## Scoring Criteria

- **0.9-1.0**: Path operations are technically sound, no redundant steps, clear logical flow
- **0.7-0.8**: Path is basically reasonable, with 1-2 minor redundancies but maintains good technical quality
- **0.5-0.6**: Path is technically feasible but has obvious redundancies or inefficient operation sequences
- **0.3-0.4**: Path has many technical issues, with obvious logical errors or invalid operations
- **0.0-0.2**: Path is severely redundant or contains erroneous operations that may cause execution failures

## Output Format

Quality Score: [numerical value from 0.0-1.0]

## Important Considerations

- Focus on technical correctness and operation efficiency of the path itself
- Consider real user operation habits and interface response characteristics
- Distinguish between necessary waiting/confirmation operations and actual redundant steps
- Evaluate path quality based solely on the operation sequence and GUI state transitions

**User**

Actions: [all_history_actions]

[current_screenshot]

**Assistant**

Quality Score: [numerical value from 0.0-1.0]

Figure 15: Prompt for intent recycling filter.

**CAGUI** Zhang et al. (2025b) is a Chinese benchmark dataset covering both grounding and agent tasks in realistic mobile applications. It consists of 600 tasks across a variety of app scenarios, with each task comprising a natural-language query, a screenshot, and the corresponding answer operation, totaling 4,516 single-step images. CAGUI enables robust evaluation in multilingual settings and provides a comprehensive assessment of model reliability and adaptability to unseen tasks and interfaces. Furthermore, since its training set has not been released and all 600 tasks are used for testing, CAGUI is particularly suitable for evaluating the generalization ability of GUI agents in novel scenarios. In our experiments, we use CAGUI for scenario generalization testing to thoroughly assess the robustness and adaptability of the method.

---

**Prompt for Intent Generation**

**System Message**

# Intent Generation Expert

You are an intent generation expert. Given a sequence of GUI operations and corresponding UI screenshots provided by the user, your task is to generate a concise intent description that represents the task or request that could be accomplished through this operation path.

## Task Overview

- **Input**: a sequence of GUI operations along with the post-operation screenshot
- **Output**: A clear, actionable intent description written as a command or request

## Intent Generation Guidelines

### 1. Analysis Focus

- **Operation Pattern Recognition**: Identify the overall behavior pattern from the sequence of actions
- **UI State Progression**: Analyze how the interface states change throughout the path
- **Completion Indicators**: Determine what the user has actually achieved by the end of the sequence

### 2. Intent Characteristics

- **Actionable**: The intent should describe a concrete task or goal as a command
- **Complete**: Reflect the full scope of what can be accomplished through this path
- **Natural**: Use language that resembles a natural user request or instruction
- **Specific**: Include relevant details (app names, search terms, specific actions) when they add clarity

### 3. Description Standards

- **Perspective**: Use imperative form as commands or requests ("Open...", "Search for...", "Add...")
- **Tense**: Use present tense imperative to indicate actionable requests
- **Conciseness**: One clear sentence that captures the essence of the task
- **Specificity**: Include key details like app names, search queries, or specific items when relevant

## Output Requirements

**Format**: JSON only, no additional text or explanations

**Structure**:

```json
{"description": "A single sentence describing the task as a command or request"}
```

**User**

Actions: [all_history_actions]

[current_screenshot]

**Assistant**

Description : [A single sentence describing the task as a command or request]

---

Figure 16: Prompt for intent generation of recycling strategy.

## B.4 METRICS

We evaluate model performance using several action-level and type-specific metrics. The main definitions are as follows:

**Step Success Rate (SR)** The proportion of steps where the predicted action type and all corresponding parameters exactly match the ground truth:

$$\text{SR} = \frac{\sum_{i=1}^{N} \mathbb{I}(\text{exact\_match}_i)}{N} \tag{4}$$

where $N$ is the total number of steps and $\mathbb{I}(\cdot)$ is the indicator function.

**Action Type Accuracy (TP)** The proportion of steps where the predicted action type matches the ground truth, regardless of parameters:

$$\text{TP} = \frac{\sum_{i=1}^{N} \mathbb{I}(\text{type\_match}_i)}{N} \tag{5}$$

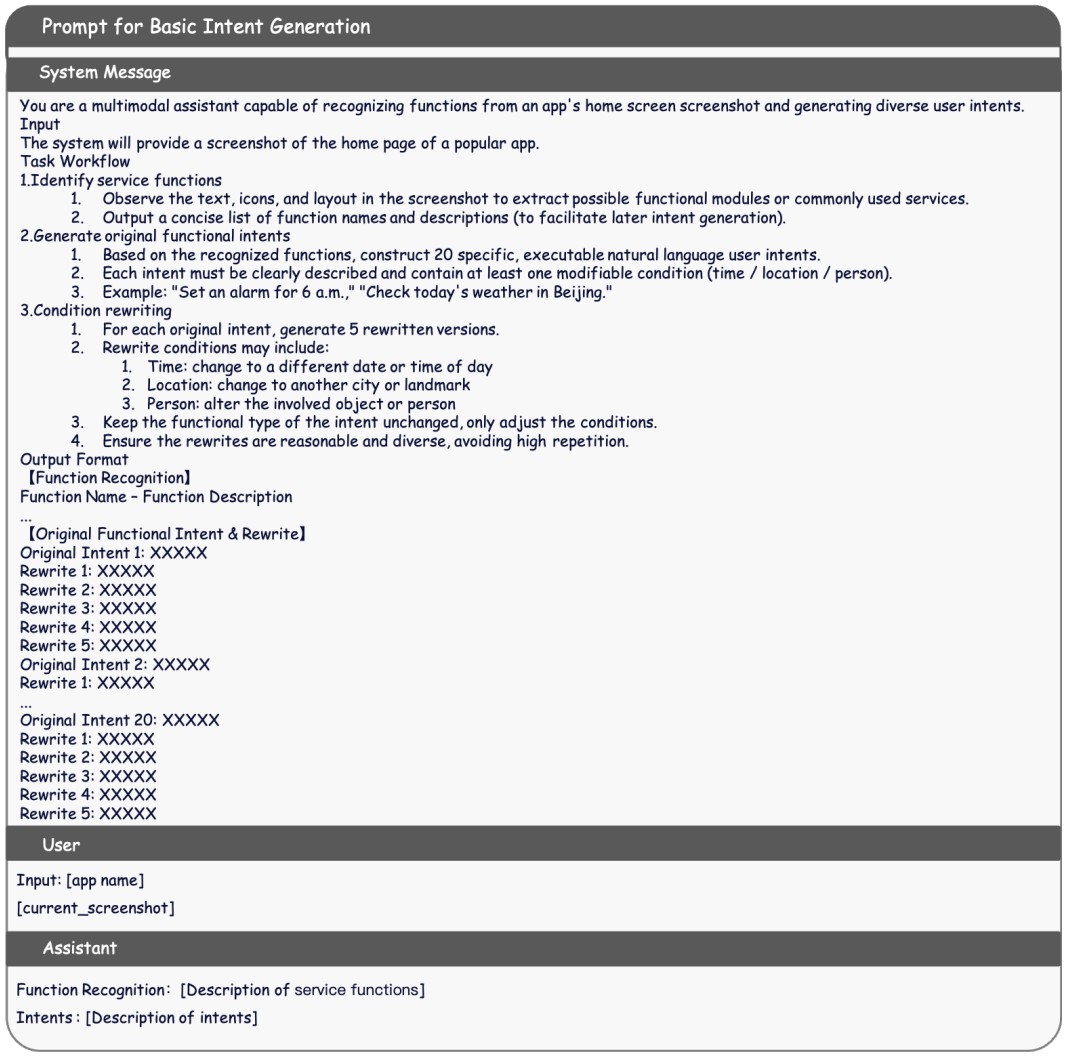

**Prompt for Basic Intent Generation**

**System Message**

You are a multimodal assistant capable of recognizing functions from an app's home screen screenshot and generating diverse user intents.
Input
The system will provide a screenshot of the home page of a popular app.
Task Workflow
1.Identify service functions
    1.   Observe the text, icons, and layout in the screenshot to extract possible functional modules or commonly used services.
    2.   Output a concise list of function names and descriptions (to facilitate later intent generation).
2.Generate original functional intents
    1.   Based on the recognized functions, construct 20 specific, executable natural language user intents.
    2.   Each intent must be clearly described and contain at least one modifiable condition (time / location / person).
    3.   Example: "Set an alarm for 6 a.m.," "Check today's weather in Beijing."
3.Condition rewriting
    1.   For each original intent, generate 5 rewritten versions.
    2.   Rewrite conditions may include:
        1.   Time: change to a different date or time of day
        2.   Location: change to another city or landmark
        3.   Person: alter the involved object or person
    3.   Keep the functional type of the intent unchanged, only adjust the conditions.
    4.   Ensure the rewrites are reasonable and diverse, avoiding high repetition.
Output Format
【Function Recognition】
Function Name – Function Description
...
【Original Functional Intent & Rewrite】
Original Intent 1: XXXXX
Rewrite 1: XXXXX
Rewrite 2: XXXXX
Rewrite 3: XXXXX
Rewrite 4: XXXXX
Rewrite 5: XXXXX
Original Intent 2: XXXXX
Rewrite 1: XXXXX
...
Original Intent 20: XXXXX
Rewrite 1: XXXXX
Rewrite 2: XXXXX
Rewrite 3: XXXXX
Rewrite 4: XXXXX
Rewrite 5: XXXXX

**User**

Input: [app name]

[current_screenshot]

**Assistant**

Function Recognition: [Description of service functions]

Intents: [Description of intents]

Figure 17: Prompt for basic intent generation.

**Mining Success Ratio (MSR)** The proportion of successful mining attempts, where a mining task is considered successful if the model judges it as successful and the number of mining steps does not exceed the predefined upper limit; otherwise, it is counted as a failure:

$$\text{MSR} = \frac{N_{\text{mining\_success}}}{N_{\text{mining\_total}}} \tag{6}$$

where $N_{\text{mining\_success}}$ denotes the number of successful mining attempts, and $N_{\text{mining\_total}}$ is the total number of mining attempts.

**Data Quality Accuracy (DQA)** The proportion of trajectories that are entirely correct and all trajectories. A trajectory is considered correct if it aligns with the expected intent, executes all actions without errors, and successfully completes the instruction; otherwise, it is counted as incorrect:

$$\text{DQA} = \frac{N_{\text{correct}}}{N_{\text{total}}} \tag{7}$$

where $N_{\text{correct}}$ is the number of correct trajectories, and $N_{\text{total}}$ is the total number of trajectories.

All metrics are computed using strict matching rules as described in our implementation. For action-specific accuracy, only steps corresponding to the respective action type are included in the denominator.

Figure 18: Prompt for complex intent generation.

### B.5 BASELINES

We compare our method with the following baselines to demonstrate its superiority. The closed-source baselines include major MLLMs such as GPT-4o Hurst et al. (2024), Gemini 2.0 Deepmind (2024) and Claude Anthropic (2024). For open-source models, we include OdysseyAgent Lu et al. (2025a), SpiritSight-8B Huang et al. (2025), GUI-R1-7B Luo et al. (2025), OS-Atlas-7B Wu et al. (2025a), Aguvis-7B Xu et al. (2025b), InfiGUI-R1-3B Liu et al. (2025), Qwen2.5-VL-3B Bai et al. (2025) and Qwen2.5-VL-7B Bai et al. (2025). We further consider representative models trained on automatically mined data, such as OS-Genesis-7B Sun et al. (2025) and GUI-Owl-7B Ye et al. (2025), as well as the model based on private human-annotated data, *i.e.*, UI-TARS-7B Qin et al. (2025). Baseline results and evaluation protocols follow the original paper to ensure fairness.

### B.6 CALCULATION OF DATASET CONSTRUCTION COST

The costs for the other datasets are calculated using the following formula:

$$C_{\text{other}} = N_{\text{img}} \times (t_{\text{annot}} + t_{\text{inspect}}) \times r_{\text{wage}}, \tag{8}$$

where $N_{\text{img}}$ refers to the images number. The total cost to construct our dataset is \$466, which is composed of \$196 for manual quality inspection and \$270 for computational overhead, as derived from the following formula:

$$C_{\text{total}} = C_{\text{inspect}} + C_{\text{comp}}, \quad C_{\text{inspect}} = N_{\text{img}} \times t_{\text{inspect}} \times r_{\text{wage}}, \tag{9}$$

$$C_{\text{comp}} = T_{\text{train}} \times G_{\text{train}} \times p_{\text{train}} + T_{\text{mine}} \times G_{\text{mine}} \times p_{\text{mine}}, \tag{10}$$

where $N_{\text{img}} = 20{,}000$ is the number of images. The training stage runs for $T_{\text{train}} = 24\,\text{h}$ on $G_{\text{train}} = 8$ A100-80G GPUs at a rental price of $p_{\text{train}} = \$0.924$/GPU/h. The data mining stage runs for $T_{\text{mine}} = 26.7\,\text{h}$ on $G_{\text{mine}} = 8$ L40-45G GPUs, priced at $p_{\text{mine}} = \$0.433$/GPU/h, based on Vast.ai Vast.ai (2025).

### B.7 INFRASTRUCTURE FRAMEWORK AND ENVIRONMENT INTERACTION

As illustrated in Fig. 4 of the main paper, our infrastructure framework adopts a layered architecture. The data layer organizes intents, interaction trajectories, screenshots, and metadata via an intent–trajectory tree structure. The engine layer implements a progressive training framework that starts with pre-training and then goes through three subsequent stages (Stage 1 to Stage 3), enabling support for a model-in-the-loop training strategy. The algorithm layer centers on a Monte Carlo Tree Search (MCTS) controller to handle selection, expansion, simulation, and backpropagation. The agent layer introduces multi-agent collaboration mechanisms for inference, orchestration, and reward evaluation during data mining. The execution layer integrates action parsing with toolchains, translating text/action parsing into ADB interface calls such as click, type, and swipe. The environment layer provides sandbox and virtual machine cluster support including environment control, isolation and security, and resource monitoring.

We use the Android Studio virtual machine (Android API 36) as our sandbox environment. We use adb commands to capture GUI screenshots from the virtual machine. To control the virtual machine, we parse the structured output (i.e., JSON code) of the GUI Agent into executable adb commands. In each iteration of the mining algorithm, we first obtain a screenshot of the virtual machine via adb commands. The Python process then retrieves the screenshot and combines it with a text prompt to construct the prompt. This prompt is fed to the inferAgent, which then produces a structured representation (i.e., JSON code) of an action. We parse the structured representation into the corresponding adb command, send the command to the virtual machine, and the virtual machine executes the operation accordingly. This completes one cycle of *state acquisition* → *action reasoning* → *operation execution*.

## C VISUALIZATION RESULTS

### C.1 EXAMPLES OF ERRONEOUS SAMPLES IN PUBLIC DATASETS

In this section, we present erroneous samples of AC and AITZ. As shown in Fig. 19,

### C.2 EXAMPLES OF INTENT-TRAJECTORY TREE

Examples of intent-trajectory trees for the AITZ and AndroidControl datasets are presented in Fig. 20 and Fig. 21, respectively. Each intent trajectory tree contains multiple intent-trajectory pairs. Taking Fig. 20 as an example, there are three intent-trajectory pairs in total. The original intent is "search for usb-a to usb-c on amazon.com then add to cart", while the recycled intents are "Search for smart watches in the Amazon app" and "Search for USB-A to USB-C cables on Amazon". Based on our approach, we are able to generate diverse potential intents, thereby achieving significant diversity beyond the constructed intents.

### C.3 EXAMPLES OF DATA MINING PROCESS

We present a **demo video** of the data mining process using $M^2$-Miner on both AndroidControl and CAGUI benchmark suites. As shown in the video, our method is able to efficiently perform intent trajectory mining across both Chinese and English app scenarios. This highlights the effectiveness and versatility of $M^2$-Miner in handling diverse language settings and application domains.

### C.4 EXAMPLES OF GUI AGENT INFER

In this section, we present testing cases of $M^2$-Miner-7B on three representative benchmark datasets: AndroidControl, AITZ and CAGUI. As shown in Fig. 22, 23, we visualize all sequential images contained within an intent for AndroidControl and AITZ. For each image, we provide the intent, thought, exact_match, and action results, where the predicted action is marked with a red triangle and the ground truth action is marked with a green circle. In addition, we present a more intuitive example of CAGUI in the **demo video**. These visualizations demonstrate that the GUI agent trained with our method can generalize well across different environments and task lengths, achieving promising results in real-world scenarios.

Intent: How do I get to the nearest AT&T Store? (AITZ)

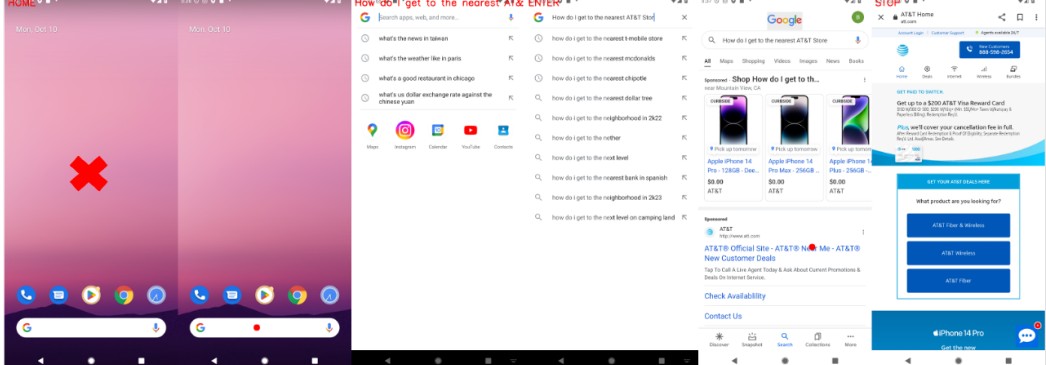

Intent: My phone is not functioning properly, therefore I'm going to turn it on service. Upload the DIY project file to Google Drive to save it. (AC)

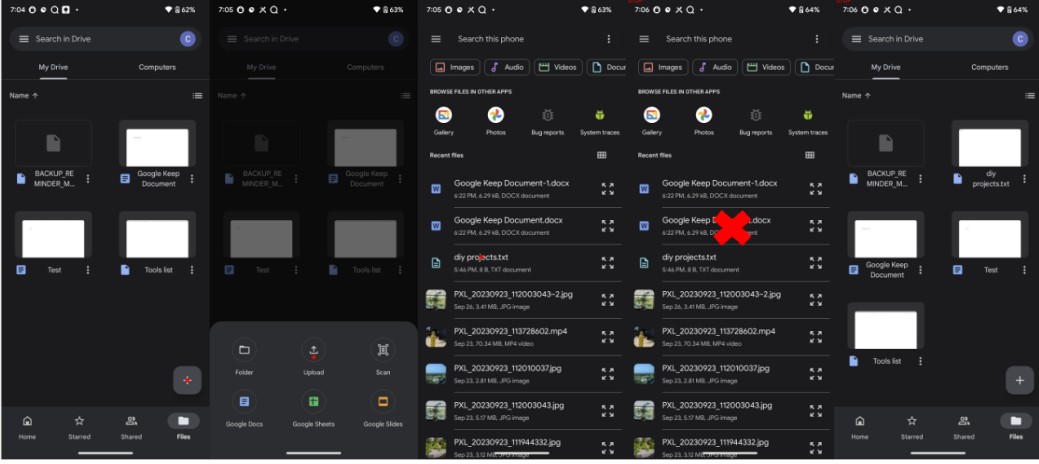

Figure 19: Erroneous samples of AC and AITZ.

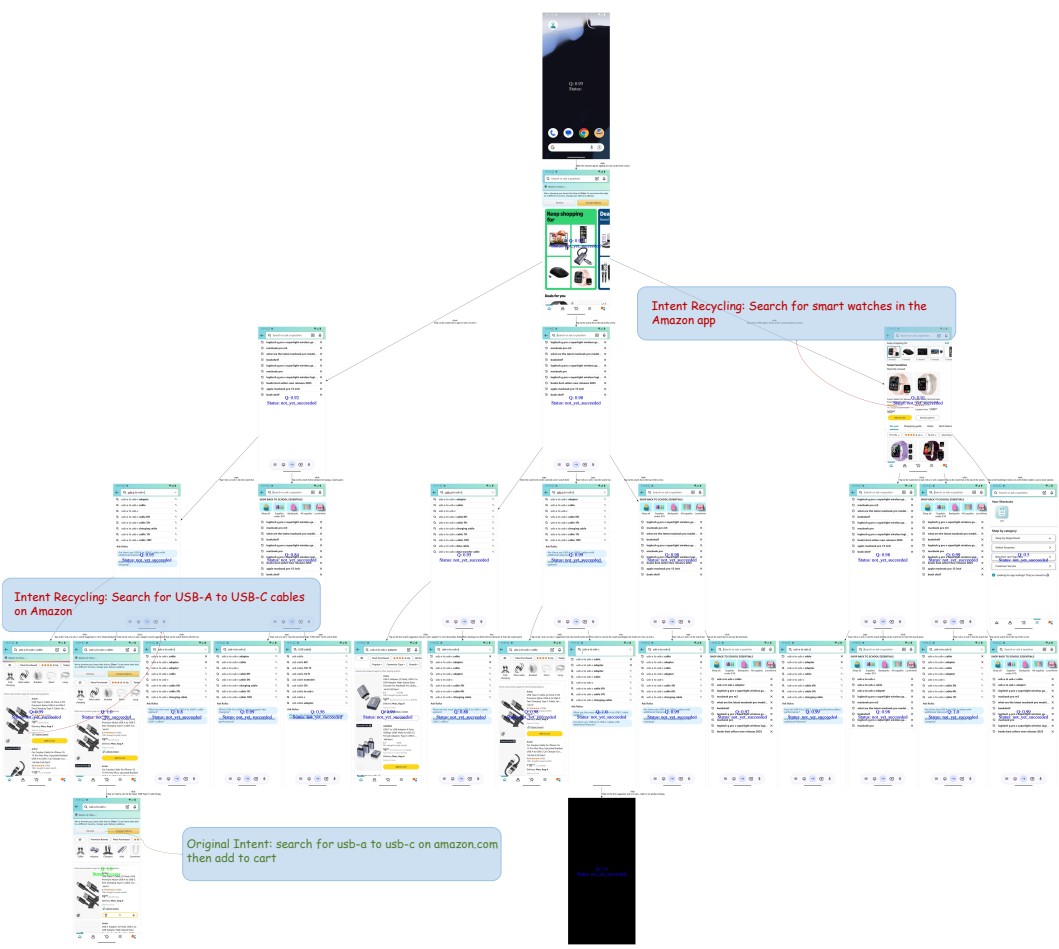

Figure 20: Example of Intent-Trajectory Tree on AITZ. The number of candidate actions $K$ is set to 3. The original intent is marked in green, while recycled intents are marked in red. Two intent-trajectory data are additionally recycled through our method.

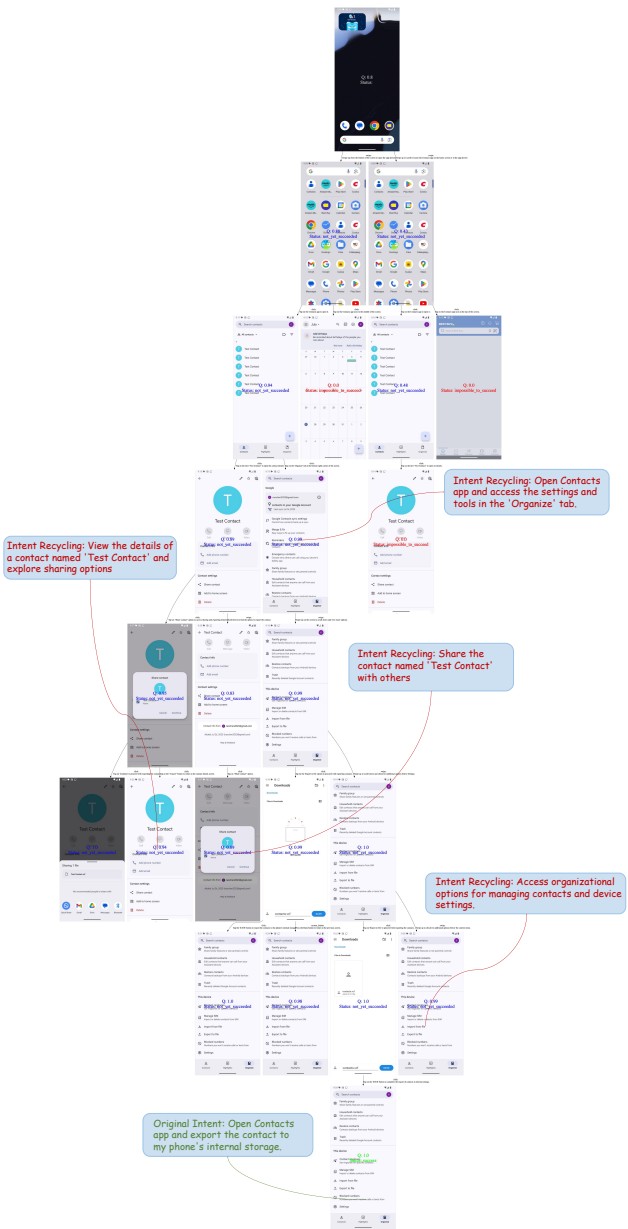

Figure 21: Example of Intent-Trajectory Tree on AndroidControl. The number of candidate actions $K$ is set to 2. The original intent is marked in green, while recycled intents are marked in red. Two intent-trajectory data are additionally recycled through our method.

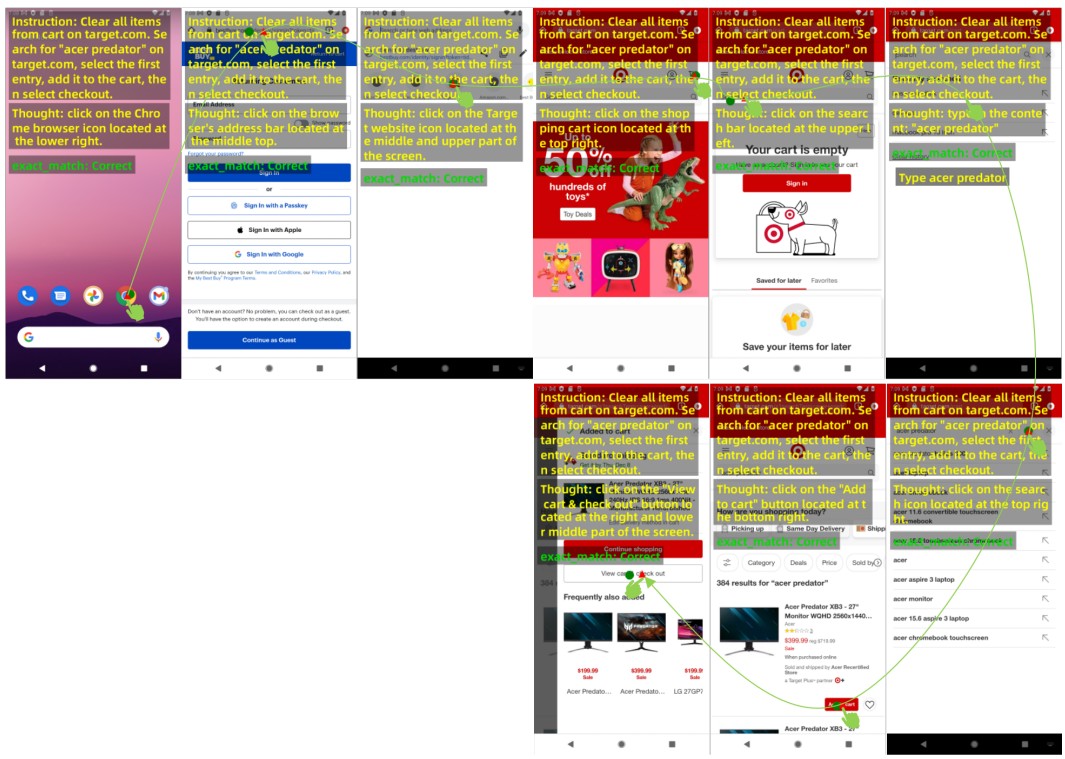

Figure 22: Test Case on AITZ. The user intent is "Empty the shopping cart on bestbuy.com. Search for razer kraken on bestbuy.com, select the first entry, add it to the cart, then select checkout."

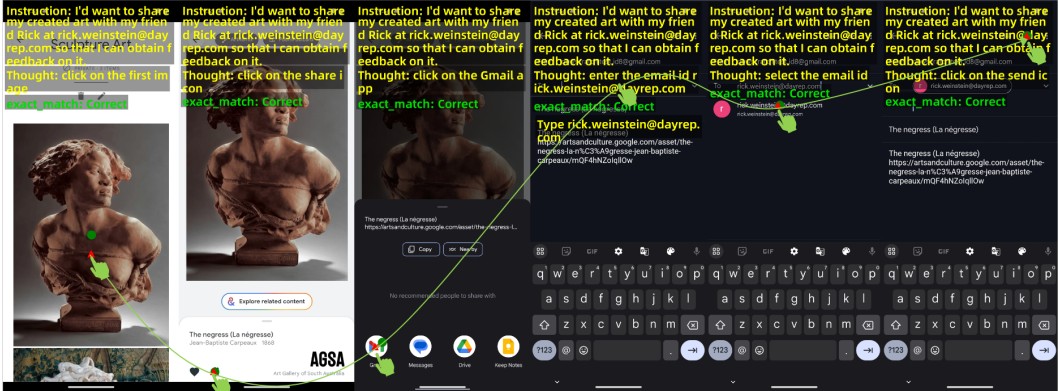

Figure 23: Test Case on AndroidControl. The user intent is "I'd want to share my created art to my friend Rick at rick.weinstein@dayrep.com so that I can obtain feedback on it."

