# OpenReview forum: "M$^2$-Miner: Multi-Agent Enhanced MCTS for Mobile GUI Agent Data Mining"
_ICLR.cc/2026/Conference — ICLR 2026 Poster_

### Official Review · Reviewer_gnp5 · 2025-10-18

**Soundness:** 2
**Presentation:** 3
**Contribution:** 3
**Rating:** 4
**Confidence:** 4

**Summary:**

The paper introduces M-Miner, a low-cost and automated framework for mining high-quality intent–trajectory pairs to train mobile GUI agents. It builds on Monte Carlo Tree Search and uses a collaborative multi-agent setup with InferAgent for exploration guidance, OrchestraAgent for coordination and speedup, and JudgeAgent for trajectory evaluation. An intent recycling strategy increases intent diversity, and a progressive model-in-the-loop scheme improves mining success over time. Experiments indicate that agents fine-tuned on the mined data achieve state-of-the-art on several mobile GUI benchmarks.

**Strengths:**

1. The M-Miner multi-agent framework automatically collects data based on intents and appears to be the first work to do so; it reduces the time and cost of manual annotation.
2. Models trained on this dataset achieve strong performance, indicating the dataset’s potential value.
3. The framework demonstrates sufficient novelty, and ablation experiments show that each component plays a sufficient role.
4. The paper is clearly structured and easy to understand.

**Weaknesses:**

1. The chapter organization needs adjustment. The theoretical preliminaries of MCTS are not difficult for researchers in RL-based agent domains. The authors devote about a page to this, which seems unnecessary. It could be moved to the appendix, freeing space to detail the three-stage intent component and the training procedure. A table-style exposition would suffice rather than a full appendix-level example.
2. Although the results are SOTA, the gains are quite limited relative to some Qwen2-VL–based methods; see the Questions for specifics.
3. The approach assumes expansion from an existing dataset rather than starting entirely from scratch. It is unclear why data were not collected from scratch on the same apps. As presented, the method resembles data augmentation.
4. The quality validation reveals shortcomings in the method, and the paper does not discuss the cost of human quality review.

**Questions:**

1. I think the comparison in Table 1 is unfair and lacks practical significance. Were the time costs considered? Table 2 reveals limitations of the method. How exactly is the human quality review conducted? And its cost?
2. In Figure 3, the content measured by the acc metric is not defined.
3. Line 373, the explanation of CAGUI may be problematic. Since the training data are not released, how do you ensure there is no test-data leakage during training? Generalization experiments are generally conducted entirely on unseen apps.
4. Baselines are missing, e.g., [1] Falcon-UI: Understanding GUI Before Following User Instructions and [2] MobileIPL: Enhancing Mobile Agents’ Thinking Process via Iterative Preference Learning. To my knowledge, these are not concurrent works.
5. Table 1 mentions AMEX and GUI-Odyssey as standard GUI evaluation datasets. I recommend adding experiments on these datasets (not strictly required during the rebuttal phase, given time constraints).
6. In Section 4.4, the intent evolution procedure appears not very different from the instruction filtering in [3] DiGIRL: Training in-the-wild Device-control Agents with Autonomous Reinforcement Learning. Is this understanding correct?
7. M2-Miner-Agent training dataset + other datasets like AITZ do demonstrate the validity of the data, but how can we ensure that there is no training data leakage, because the newly generated instructions are very similar to the original ones, but we all know that the original train and test instructions are also very similar and share the same data distribution.
8. If the original data is removed from the ablation experiment and only the mined data is used, what will the model results be?
9. What is the difference between mining data from scratch using brand new instruction templates in your framework, compared to replacing existing instructions with slots? I think a comparison might be necessary, as the paper doesn't explain why it's necessary to mine data on existing data.

---

> ### Author Response · Authors · 2025-11-22
> **Response to Reviewer gnp5 (Part 1/3)**
>
> We are very glad that you recognize the significance and sufficient novelty of our work. We appreciate your detailed feedback and suggestions for improvement. We treasure the opportunity to address your concerns and improve our work.
>
> ---
>
> **W1: The chapter organization needs adjustment.**
>
> We have moved the preliminaries of MCTS to the appendix A.1 and provided detailed explanations of the three-stage intent component and the training procedure in Section 3.4 (revised manuscript), along with illustrative diagrams for more intuitive understanding. Additionally, at the beginning of the appendix, we provide an outline for each section to enhance readability.
>
> ---
>
> **W2 & Q4: Concerns regarding missing baselines and the associated performance gains.**
>
> Following the reviewer's suggestion, we have incorporated comparisons with these two Qwen2-VL-based baselines, Falcon-UI and MobileIPL. The results are shown in the table below. As the new results indicate, while our method achieves modest improvements on the AC-High and AITZ benchmarks, it demonstrates a significant performance gain on AC-Low. In addition, we further conducted comprehensive comparisons with three concurrent Qwen2.5-VL-based methods (GUI-R1-7B, InfiGUI-R1, and GUI-Owl-7B) across more benchmarks such as GUI-Odyssey and CAGUI. The results demonstrate that the SR of our method outperforms all Qwen2.5-VL based approaches on various benchmarks. However, because the models of the two Qwen2-VL-based baselines are not publicly available, we were unable to perform direct comparisons on these datasets.
>
> We appreciate the reviewer's focus on performance metrics. Beyond the numerical improvements, we wish to respectfully emphasize that the core contribution of our work is the automated data production framework itself, which we believe holds significant practical value for the GUI Agent domain and could enhance the GUI Agent's execution performance on new GUI environments from a data-centric perspective, without the need for manual annotation.
>
> | Models              | **AC-Low** | **AC-Low** | **AC-High** | **AC-High** | **AITZ** | **AITZ** | **GUI-Odyssey** | **GUI-Odyssey** | **CAGUI** | **CAGUI** |
> | ------------------- | :-------: | :----: | :---------: | :----: | :------: | :----: | :-------------: | :----: | :-------: | :----: |
> |                     | TP        | SR     | TP          | SR     | TP       | SR     | TP              | SR     | TP        | SR     |
> | *Qwen2-VL Based Models* |         |        |             |        |          |        |                 |        |           |        |
> | Falcon-UI           | -         | 86.6   | -           | 72.7   | **84.7** | 69.1   | -               | -      | -         | -      |
> | MobileIPL           | -         | 77.0   | -           | 72.7   | _81.9_ | _69.2_ | -             | -      | -         | -      |
> | *Qwen2.5-VL Based Models* |         |        |             |        |          |        |                 |        |           |        |
> | GUI-R1-7B           | 85.2      | 66.5   | 71.6         | 51.7   | 56.8     | 50.5   | 65.5            | 38.8   | -         | -      |
> | InfiGUI-R1          | 96.0      | 92.1   |  **82.7**     | 71.1   | 70.7     | 52.9   | 74.5            | 55.0   | -         | -      |
> | GUI-Owl-7B          | 93.8      | 90.0   | 81.5         | _72.8_ | 78.9   | 65.1   | 83.4            | 60.7   | 80.0      | 59.2   |
> | **M²-Miner-3B**     | _97.2_  | _93.2_ | 81.3       | 71.2   | 78.6     | 66.6   | _88.2_        | _77.1_ | _88.5_ | _67.3_ |
> | **M²-Miner-7B**     | **97.5**  | **93.5** | _81.8_ | **72.9** | 81.3   | **69.4** | **90.5**        | **79.3** | **88.8** | **70.2** |
>
> ---
>
> **W3 & Q9: Why data were not collected from scratch on the same apps? Why mine data on existing data?**
>
> We would like to clarify that our approach collect the data entirely from scratch as stated on line 309 of our first manuscript (*i.e.,* line 313 of our revised manuscript), rather than augmenting or extending existing datasets. Specifically, we first collect the home screen screenshots of targeted apps, and then, summarize the commonly used services of the application and generate user intents based on MLLM. Subsequently, the slots of these generated intents are modified to perform intent expansion. The entire intent generation process is independent of any existing datasets. The Prompt for basic intent generation is shown in Figure 17.
> Moreover, for a more intuitive visualization of the intent construction process, we have incorporated an illustrative figure (Figure 3) into Section 3.4 of the revised manuscript.

---

> ### Author Response · Authors · 2025-11-22
> **Response to Reviewer gnp5 (Part 2/3)**
>
> **W4 & Q1: Concerns regarding the cost of human quality review.**
>
> We have addressed this concern in the **"Cost Calculation and Comparison"** part of our General Response.
>
> ---
>
> **Q2: Acc metric of Figure 3 is not defined.**
>
> Thank you for your valuable suggestion. "Acc'' metric of Figure 3 in our first manuscript refers to Data Quality Accuracy (DQA), which evaluates the correctness of the collected trajectories. We define DQA as follows:
>
> \begin{equation}
> DQA = \frac{N\_{\text{correct}}}{N\_{\text{total}}},
> \end{equation}
>
> where $N_{correct}$ is the number of correct trajectories, and $N_{total}$ is the total number of trajectories. A trajectory is considered correct if it aligns with the expected intent, executes all actions without errors, and successfully completes the instruction.
> In our new manuscript, we update Figure 5 label from "Acc'' to "DQA'' to avoid ambiguity. Furthermore, we have clarified the definition of DQA and its calculation method in Appendix B.4.
>
> ---
>
> **Q3: Explanation of CAGUI at line 373 may be problematic.**
>
> We agree with the reviewer that the generalization evaluation for GUI agents should be conducted entirely on unseen applications. Regarding the experiments on CAGUI, we would like to clarify the following.
>
> The aim of this experiment is twofold: (1) to verify the **generalization ability of our framework** in mining high-quality data in new scenarios; and (2) to assess the **effectiveness of the mined data** in improving model performance. The framework’s input consists solely of the applications name and their home pages. The mining process strictly follows the procedure shown in Figure 3 to generate intents, with the intent source completely independent of the original dataset to ensure strict data isolation. As shown in Figure 8 of the revised manuscript, the mining framework achieves an MSR of 46.8% in new scenarios, demonstrating its capability to successfully mine data from unseen applications. Furthermore, training models with the mined data leads to an additional SR improvement of 5.7%, confirming their effectiveness. Since the entire process is conducted under strict data isolation, these improvements can be attributed to the framework’s data production capability rather than any form of data leakage.
>
> Accordingly, we have revised the statements in the manuscript to ensure greater rigour and clarity.
>
> ---
>
> **Q5: Adding experiments on AMEX and GUI-Odyssey datasets.**
>
> We have added experiments on GUI-Odyssey, and the results are presented in the updated Table 2 of our new manuscript, which indicate that our method still achieves SOTA performance on this dataset. We should clarify that for the evaluation on this dataset, the model was trained exclusively on the prior data. We did not perform any data mining specifically for the apps within this dataset, due to time constraints.
> Regarding the AMEX dataset, we did not use it for benchmarking for two primary reasons. First, the official code does not provide a standard test set split. Second, the authors of AMEX themselves stated in GitHub that *"We decided not to use it as a benchmark but a pure supportive and supplementary dataset for element grounding and instructive operation."*
>
> ---
>
> **Q6: Is there a significant difference between the intent evolution procedure in Section 4.4 and the instruction filtering in DiGIRL?**
>
> The two concepts are fundamentally different. DiGIRL focuses on filtering existing data, whereas our method centers on generating new intents.
> DiGIRL trains two value functions to filter existing intent–trajectory data based on the difficulty of the intents, and then uses the filtered data to train the actor, as illustrated in Figure 5 of their paper.
> In contrast, our intent evolution process is about constructing intents. After constructing new intents, we use our mining framework to mine trajectory data specifically for these intents.

---

> ### Author Response · Authors · 2025-11-22
> **Response to Reviewer gnp5 (Part 3/3)**
>
> **Q7: Concerns regarding the training data leakage.**
>
> We thank the reviewer again for the constructive comments, which have helped us improve the quality of our manuscript. We hope the clarifications below will address the reviewer's concerns.
> Firstly, all newly generated instructions are derived from analyzing the functionalities of applications based on the homepage screenshots with MLLM. They are not sampled from the existing training or test set. We have already clarified this point in our responses to W3 & Q9.
> Besides, our work concentrates on how to automatically produce intent-trajectory data rather than on improving model performance at the algorithmic level. From a data production perspective, our mining framework, when encountering new mobile applications, analyzes their functionalities to generate potential user intents and then automatically produces GUI Agent data. This process does not involve any training data leakage.
> Beyond demonstrating that a GUI agent trained on our mined data can achieve SOTA performance on public benchmarks, the experiments in Table 2 more importantly validate the effectiveness of our mining framework. They show that our framework can successfully mine valuable intent–trajectory data, which holds practical significance for improving GUI agents’ execution performance on new mobile applications.
>
> ---
>
> **Q8: Ablation study on mined data only.**
>
> As suggested, we have conducted an ablation study on the training data, and the results are presented in Table 5 of the revised manuscript. The experiments were performed on the CAGUI benchmark using the Qwen2.5-VL-7B model.
>
>
> | Models | Training Datasets | TP | SR | CLICK | SCROLL | TYPE | PRESS | STOP |
> | --- | --- |:---:|:---:|:---:|:---:|:---:|:---:|:---:|
> | Qwen2.5-VL-7B | -- | 74.2 | 55.2 | 55.6 | 11.3 | 41.9 | 0.0 | 61.1 |
> | Qwen2.5-VL-7B | Public Datasets | 84.9 | 64.4 | 67.1 | 22.3 | 62.3 | 0.0 | 65.5 |
> | Qwen2.5-VL-7B | Auto-Mined Datasets | 87.1 | 69.5 | 70.7 | 24.7 | 68.5 | 0.0 | 78.5 |
> | Qwen2.5-VL-7B | Public and Auto-Mined Datasets | 88.8 | 70.2 | 71.2 | 26.6 | 69.5 | 0.0 | 72.8 |
>
> Without training data, the Qwen2.5-VL-7B model shows relatively low accuracy across all action types (CLICK, SCROLL, TYPE and STOP), indicating limited execution performance on new GUI applications. Training on public datasets produces moderate improvements, with SR increasing by 9.2% and performance gains observed across all action types. Using only auto-mined data results in larger boosts (SR +14.3%, CLICK +15.1%, SCROLL +13.4%, TYPE +26.6%, STOP +17.4%), highlighting the practical significance of our proposed data production framework. When combining public and auto-mined datasets, the model achieves the best results, outperforming the public-only setting by +3.9% TP, +5.8% SR, and showing consistent gains across all actions.
> In summary, while our auto-mined data serves as a valuable complement to existing public datasets, we would like to emphasize our core contribution, *i.e.,* automated data production framework, which has practical significance for the GUI Agent domain and could enhance the GUI Agent's execution performance on new GUI environments from a data-centric perspective, without the need for manual annotation.

---

> > ### Comment · Reviewer_gnp5 · 2025-11-23
> > **Official comment by Reviewer gnp5**
> >
> > Thank you for the author's reply; most of my questions have been clarified. Considering this method is data-driven, I have a further question: could the author compare this work with previous data mining methods to demonstrate its innovativeness? This is my current focus.

---

> > > ### Author Response · Authors · 2025-11-27
> > > **Response to Reviewer gnp5 (Part 1/2)**
> > >
> > > Thank you for your feedback. We appreciate the opportunity to address your concerns. Here, we provide additional analysis and clarification to address your comments.
> > >
> > > ---
> > >
> > > **Q1: Comparison of Innovativeness.**
> > >
> > > Early data mining studies were primarily designed to support LLM-based GUI agents operating in web environments, where the agents perceive the interface through the textual representation of the HTML. Driven by the rapid advancements in Vision-Language Models (VLMs) and constrained by the limited accessibility of DOM trees on mobile platforms, a substantial amount of recent GUI agent research has shifted towards purely vision-based approaches with VLMs, utilizing screenshots as the agent's observation. However, research on data mining methods tailored to such mobile GUI agents is still largely under-explored.
> > >
> > > OS-Genesis [1] is a representative work that consists of an instruction construction stage and an execution stage powered by GPT-4o. The instruction construction stage initiates by traversing interactive UI elements (e.g., via clicks) to capture single-step interface transitions. These transitions are then distilled by GPT-4o into low-level instructions (e.g., “click the dropdown to display options,” as presented in their paper). Finally, GPT-4o is utilized to generate a high-level instruction (e.g., “configure application settings,” as shown in their paper) that encapsulates the previously generated low-level instruction. Once the instructions are synthesized, OS-Genesis employs a model like GPT-4o as a GUI agent to execute them within the environment, thereby producing trajectories. Therefore, the data mining methodology of OS-Genesis is essentially to simply use a single model to execute synthetic instructions. As a result, the success of this process entirely depends on the capability of the employed task-executing model. This is consistent with the limitations discussed in their paper, where they refer to such methodology as **model-based trajectory construction**.
> > >
> > > Mobile-Agent-v3 [2] is a concurrent work that also adopts **the model-based trajectory construction paradigm** for data production. Similar to OS-Genesis, Mobile-Agent-v3 also first constructs user intents and then employs the GUI agent to obtain trajectories. This approach constructs user intents by first randomly sampling interaction paths and then utilizing a LLM to generate intents based on the metadata from these paths (e.g., detailed screenshot descriptions). In addition, compared with OS-Genesis, it further incorporates human intervention and synthetic guidance to help the GUI agent execute more accurately.

---

> > > ### Author Response · Authors · 2025-11-27
> > > **Response to Reviewer gnp5 (Part 2/2)**
> > >
> > > Compared with these model-based trajectory construction methods, we innovatively introduce Monte Carlo Tree Search into the moblie GUI agent data mining task, forming a **tree-search-based trajectory construction paradigm**, which expands the exploration space, enhances the robustness of mining, and ensures the mining results not completely dependent on the capabilities of GUI agents. However, a direct application of a vanilla MCTS algorithm for trajectory production is infeasible, due to its random expansion strategy and rollout-based reward calculation. Therefore, we innovatively propose a **collaborative multi-agent framework integrated with MCTS**. This approach exponentially improves the mining efficiency (e.g., **64× boost** at task length 9), thereby making the application of MCTS feasible. Notably, our proposed multi-agent framework differs from others designed for user intent execution. To the best of our knowledge, we are the first to introduce a multi-agent architecture specifically tailored for tree-search-based data mining, which is composed of InferAgent, OrchestraAgent, and JudgeAgent.
> > >
> > > Furthermore, our intent recycling strategy represents another novel contribution. By re-evaluating all paths within the trajectory tree, it extracts additional valuable intent-trajectory data, thereby improving mining efficiency and enriching intent diversity. Essentially, this amounts to constructing GUI agent data **based on interaction paths, rather than based on user intents** as in OS-Genesis, Mobile-Agent-v3, and our MCTS-based data mining process. Through the MCTS-based data mining process and the proposed intent recycling strategy, we obtain a multi-intent trajectory tree data structure, as shown in Figures 20 and 21 in the Appendix.
> > >
> > > In addition, the proposed model-in-the-loop training strategy improves the model's mining success rate on new mobile applications and offers a reference training approach for the GUI Agent data mining task. To this end, we build a **customized infrastructure framework** that unifies GUI agent execution, intent-trajectory data mining, and model training into an integrated pipeline, thereby forming a customized model-in-the-loop framework. The detailed description of this framework can be found in Section 4.1 and Appendix B.7 of the revised manuscript.
> > >
> > > Finally, as shown in the table below, our method consistently outperforms OS-Genesis and GUI-Owl-7B (i.e., the model from Mobile-Agent-v3) across all benchmarks.
> > >
> > > | Models              | **AC-Low** | **AC-Low** | **AC-High** | **AC-High** | **AITZ** | **AITZ** | **GUI-Odyssey** | **GUI-Odyssey** | **CAGUI** | **CAGUI** |
> > > | ------------------- | :-------: | :----: | :---------: | :----: | :------: | :----: | :-------------: | :----: | :-------: | :----: |
> > > |                     | TP        | SR     | TP          | SR     | TP       | SR     | TP              | SR     | TP        | SR     |
> > > | *Open-source Models w/ Auto-Mined Datasets* |         |        |             |        |          |        |                 |        |           |        |
> > > | OS-Genesis-7B          | 90.7      | 74.2   | 66.2     | 44.5   | 20.0     | 8.5   | 11.7            | 3.6   | 38.1         | 14.5      |
> > > | GUI-Owl-7B          | 93.8      | 90.0   | 81.5         | 72.8 | 78.9     | 65.1   | 83.4            | 60.7   | 80.0      | 59.2   |
> > > | **M²-Miner-7B**     | **97.5**  | **93.5** | **81.8**     | **72.9** | **81.3**   | **69.4** | **90.5**      | **79.3** | **88.8** | **70.2** |

---

### Official Review · Reviewer_xyHC · 2025-10-29

**Soundness:** 3
**Presentation:** 2
**Contribution:** 2
**Rating:** 6
**Confidence:** 2

**Summary:**

This paper propose M2-Miner, the first low-cost and automated mobile GUI agent data-mining framework based on Monte Carlo Tree Search(MCTS). For better data mining efficiency and quality, the paper present a collaborative multi-agent frame work, comprising InferAgent, OrchestraAgent, and JudgeAgent for guidance, acceleration, and evaluation. To further enhance the efficiency of mining and enrich intent diversity, this paper design an intent recycling strategy to extract extra valuable interaction trajectories. Additionally, a progressive model-in-the-loop training strategy is introduced to improve the success rate of data mining.

**Strengths:**

1.This paper propose a fully automated framework for mobile GUI agent data mining. By introducing MCTS and designing a collaborative multi-agent framework, the method improve data mining efficiency while enhancing data quality.
2.The intent recycling strategy further enhances both mining efficiency and intent richness, while the progressive model-in-the-loop training paradigm boosts success rates in both familiar and novel environments.
3.Extensive experiments show that GUI agents trained on the mined data achieve SOTA performance.

**Weaknesses:**

1. The paper propose an automated mobile GUI agent data-mining framework based on Monte Carlo Tree Search(MCTS). Monte Carlo tree search is a classic algorithm, is its innovation insufficient?
2.The background knowledge of MOBILE GUI AGENT DATA MING was not sufficiently introduced in the paper writing, making it difficult to understand.

**Questions:**

see the weaknesses

---

> ### Author Response · Authors · 2025-11-22
> **Response to Reviewer xyHC**
>
> We appreciate that the reviewer recognizes the four contributions of our work. We are grateful for your positive feedback and suggestions for improvement. We treasure the opportunity to address your concerns and improve our work.
>
> ---
>
> **Background Knowledge of Mobile GUI Agent Data Mining.**
>
> A Mobile GUI Agent refers to a system that utilizes LLMs or MLLMs to operate mobile devices on behalf of a user, executing given instructions. Training such an agent requires a vast amount of intent-trajectory pairs, but this type of interaction data remains scarce.
>
> Moreover, existing GUI agent datasets, such as AITZ [1], AndroidControl [2] and GUI‑Odyssey [3], are entirely manually constructed. This manual annotation process is very costly, demanding specialized annotation tools and experienced annotators.
>
> Mobile GUI Agent Data Mining is the process of automatically collecting and labeling interaction trajectories using an automated framework, which can also be called Mobile GUI Agent Data Production (or Generation).
>
> While the majority of current research focuses on improving model performance at the algorithmic level using existing open-source data, we approach this challenge from a different perspective. Our work concentrates on how to automatically produce intent-trajectory data, thereby reducing the cost of manual annotation. By doing so, we aim to enhance the agent's execution performance on new mobile applications from a data-centric standpoint.
>
> ---
>
> **W1: Concerns about insufficient innovation.**
>
> Our work focuses on the challenge of automatically producing intent-trajectory data. This is crucial for reducing the cost of manual annotation and for enhancing the model's execution performance on new mobile applications from a data-centric perspective.
>
> Notably, we are the first to introduce Monte Carlo Tree Search (MCTS) into the task of mobile GUI agent data mining. However, a direct application of a vanilla MCTS algorithm for trajectory production is infeasible, due to its random expansion strategy and rollout-based reward calculation.
>
> Therefore, we innovatively propose a collaborative multi-agent framework integrated with MCTS. This approach exponentially improves the mining efficiency (*e.g.,* 64× boost at task length 9), thereby making the application of MCTS feasible.
>
> Furthermore, our novel intent recycling strategy is also a noteworthy design. It enhances both mining efficiency and intent diversity by extracting additional valuable interaction trajectories from the search tree. In addition, the proposed model-in-the-loop training strategy improves the model's mining success rate on new mobile applications and offers a reference training approach for the GUI Agent data mining task.
>
> ---
>
> **W2: The background knowledge of mobile GUI agent data mining was not sufficiently introduced.**
>
> We greatly appreciate this valuable feedback. In the revised version of our submission, we have supplemented the *Related Works* section with background knowledge on GUI Agent Data Mining and renamed the corresponding subheading to *GUI Agent Data Production*.
>
> ---
>
> **References**
>
> [1] Li, Wei, et al. "On the effects of data scale on ui control agents." Advances in Neural Information Processing Systems 37 (2024): 92130-92154.
> [2] Zhang, Jiwen, et al. "Android in the zoo: Chain-of-action-thought for gui agents." Findings of the Association for Computational Linguistics: EMNLP 2024. 2024.
> [3] Lu, Quanfeng, et al. "GUIOdyssey: A Comprehensive Dataset for Cross-App GUI Navigation on Mobile Devices." Proceedings of the IEEE/CVF International Conference on Computer Vision. 2025.

---

> > ### Comment · Reviewer_xyHC · 2025-11-24
> > **comment**
> >
> > Thank you for the author's response. The reviewer has not conducted any relevant research in the field of Mobile GUI Agents.
> > The reviewer will maintain the current score.

---

> > > ### Author Response · Authors · 2025-11-27
> > > **Official Comment by Authors**
> > >
> > > Thank you for your thoughtful feedback and active engagement throughout the rebuttal process. We sincerely appreciate your support and endorsement!

---

### Official Review · Reviewer_eLqw · 2025-10-29

**Soundness:** 3
**Presentation:** 2
**Contribution:** 3
**Rating:** 4
**Confidence:** 3

**Summary:**

This paper proposes a multi-agent framework for mobile GUI agents. By introducing a tree-based structure, it organizes and stores learned operations in a structured and reusable manner. The framework incorporates three specialized agents to optimize the MCTS search process and reward estimation, effectively avoiding inefficient random exploration. Furthermore, it introduces a model-in-the-loop training paradigm that enables continual learning and self-improvement during deployment, expanding the system’s learning capability from a relatively small initial dataset. Experimental results show significant performance improvements, and comprehensive ablation studies validate the effectiveness of each component.

**Strengths:**

S1. Strong Experiments. It compares with 13 methods and analyzes the effect of agent numbers and online learning strategies, showing solid and comprehensive evaluation.

S2. Practical Significance. The framework is scalable and adaptable, demonstrating potential for real-world GUI automation and broader mobile applications.

S3. Trajectory Recycling is an interesting and computation-efficient design.

**Weaknesses:**

W1. Writing and Presentation Issues. The paper contains several typos and minor writing problems that affect readability:

1. Line 52: “we presents” ->“we present”
2. Line 274: “where i denotes the i-th visit to the node” appears twice.
3. Line 353” “This is crucial when targeting new application scenarios.” is unclear — please specify what scenarios are referred to.
4. Line 480 “significantly improve” -> “improves”.
5. Line 484 “an solid foundation” -> “a solid foundation”
6. Line 485 “Statement of Using LLM” lacks a period at the end.

W2. Outdated Baselines. The comparison does not include the latest agent-based methods, such as AppAgent[1]. The authors should either explain the exclusion or add these newer methods to the experiments.

W3. Clarity of Agent Interaction

The description of how the three agents (Infer, Orchestra, and Judge) coordinate during the MCTS process remains vague. A clearer explanation of their information flow and role boundaries would improve reproducibility and reader understanding.

[1]. Zhang, Chi, et al. "Appagent: Multimodal agents as smartphone users." *Proceedings of the 2025 CHI Conference on Human Factors in Computing Systems*. 2025.

W4. The novelty of applying MCTS and multi-agent in the GUI data mining is unclear.

**Questions:**

Q1. Consider comparing with more recent agent-based methods, such as AppAgent [1], or provide a clear justification for why these methods were excluded.

Q2. Ablation Study. The ablation study lacks clarity on how different components are decoupled. Specifically, how does the InferAgent function when the JudgeAgent is removed? Please elaborate on how the agents’ dependencies are handled during ablation.

Q3. Experimental Clarity. It is unclear how large language models (LLMs) are wrapped or packaged as agents for GUI interaction in your experiments. Please explain how the LLMs receive GUI state inputs and produce executable actions, and whether environment feedback is included in this loop.

---

> ### Author Response · Authors · 2025-11-22
> **Response to Reviewer eLqw  (Part 1/3)**
>
> We appreciate that the reviewer recognizes the solidity of the experiments and the practical significance of our work. We thank the reviewer for the detailed feedback and constructive suggestions. Here, we treasure the opportunity to address your concerns and improve the quality of our work.
>
> ---
>
> **W1: Writing and presentation issues.**
>
> We have corrected all the writing issues mentioned and appreciate you bringing them to our attention. For the unclear sentence "This is crucial when targeting new application scenarios." at Line 353, we have rewritten the sentence for clarity. Here we mean "This is crucial when targeting new mobile applications." As for the part of "Statement of Using LLM.", we have moved it to the appendix due to space limitations.
>
> ---
>
> **W2 & Q1: Consider comparing with more recent agent-based methods, such as AppAgent.**
>
> AppAgent [1] is a pioneering work in the GUI Agent field (initially posted on arXiv in December 2023; later accepted to CHI 2025). It evaluates its method through manually set tasks rather than on publicly available benchmarks (*e.g.,* AC [2], AITZ [3], GUI Odyssey [4] and CAGUI [5]) that emerged later. We will incorporate a discussion of this work into our Related Work.
>
> Notably, many of the baselines in Table 2 are latest agent-based methods. For instance, OS-Genesis [6] (ACL 2025), OdysseyAgent [4] (ICCV 2025), and SpiritSight [7] (CVPR 2025) are all recent works from top-tier conferences. Furthermore, our comparison includes GUI-Owl-7B [8], from a very recent technical report (arXiv, Aug. 2025), and even GUI-R1 [9], a concurrent work also submitted to ICLR 2026.
>
> Additionally, to involve more recent methods, we have updated our Table 2 to include a new baseline InfiGUI-R1 [10], which is also a concurrent work submitted to ICLR 2026.
>
> ---
>
> **W3: Explain agent interaction more clearly.**
>
> We sincerely appreciate the reviewer's insightful feedback. We have modified the "Collaborative Multi-Agent Framework" section (Section 3.2 in the revised manuscript) to more clearly explain the information flow and role boundaries among the three agents.
>
> Figure 2(a) illustrates the flowchart of the MCTS algorithm integrated with our multi-agent framework. During the selection phase, the algorithm traverses the tree and selects a node to expand based on the UCT score. This phase does not involve any agents.
>
> Subsequently, in the expansion phase, the GUI screenshot of the selected node is fed to the **InferAgent**, which then generates a set of candidate actions. Figure 2(a) shows an example where two tap actions, one wait, and one swipe are generated. Note that the two tap actions are essentially equivalent. Next, the **OrchestraAgent** merges the equivalent actions and prioritizes the remaining ones. As depicted in Figure 2(a), the tap action is ultimately ranked first. This action is then executed, causing the virtual machine to transition to a new page. Consequently, the corresponding node is expanded in the search tree.
>
> The mining process then proceeds to the simulation phase. In this stage, the **JudgeAgent** analyzes the GUI screenshot of the newly expanded node, determines its status as either success, failure, or intermediate, and subsequently calculates a corresponding reward. In contrast, in the simulation phase of vanilla MCTS, a node's reward is evaluated via costly rollouts. These rollouts are executed by continuously taking forward steps until a terminal state (*e.g.,* success or failure) is reached. By replacing the outcome-based reward with a process-based reward, our method further improves the efficiency of data mining.
>
> Finally, the process enters the backpropagation phase. Based on the reward computed in the simulation phase, the values of the nodes are updated upwards along the search path, starting from the newly expanded node.
>
> These four phases are repeated in a cycle until an interaction trajectory that matches the target intent is successfully mined. Overall, our MCTS framework leverages three specialized agents: the InferAgent for action generation (expansion phase), the OrchestraAgent for merging and prioritizing actions (expansion phase), and the JudgeAgent for reward estimation (simulation phase).

---

> ### Author Response · Authors · 2025-11-22
> **Response to Reviewer eLqw (Part 2/3)**
>
> **W4: Unclear novelty of applying MCTS and multi-agent in the GUI data mining.**
>
> While the majority of current research focuses on improving model performance at the algorithmic level using existing open-source data, we approach this challenge from a different perspective. Our work concentrates on how to automatically produce intent-trajectory data. This is crucial for reducing the cost of manual annotation and for enhancing the model's execution performance on new mobile applications from a data-centric perspective.
>
> Notably, we are the **first** to introduce Monte Carlo Tree Search (MCTS) into the task of mobile GUI agent data mining. However, a direct application of a vanilla MCTS algorithm for trajectory production is infeasible, due to its random expansion strategy and rollout-based reward calculation. Therefore, we innovatively propose a **collaborative multi-agent framework** integrated with MCTS. This approach exponentially improves the mining efficiency (*e.g.,* **64×** boost at task length 9), thereby making the application of MCTS feasible.
>
> Furthermore, our novel **intent recycling strategy** is also a noteworthy design. It enhances both mining efficiency and intent diversity by extracting additional valuable interaction trajectories from the search tree. In addition, the proposed **model-in-the-loop training strategy** improves the model's mining success rate on new mobile applications and offers a reference training approach for the GUI Agent data mining task.
>
> ---
>
> **Q2: How different components are decoupled during ablation of multi-agent framework?**
>
> We have enhanced the experimental description of the ablation study for the multi‑agent framework in the revised manuscript, offering a more detailed and systematic explanation of how we isolate and evaluate the contribution of each agent to mining efficiency.
>
> For vanilla MCTS with only the **InferAgent**, the InferAgent will generate several actions during the expansion phase, but these actions are often redundant (*e.g.,* clicking the same button at different coordinates) and are not ranked by priority. The simulation phase is the same as vanilla MCTS, where node values are evaluated via costly rollouts. Rollouts are executed by continuously executing the InferAgent until it outputs success, failure, or reaches the preset maximum number of steps.
>
> When the **OrchestraAgent** is introduced in the expansion phase, it merges the redundant actions generated by the InferAgent and prioritizes correct ones, thereby reducing exploration nodes.
>
> When the **JudgeAgent** is introduced in the simulation phase, rollouts are no longer needed. Instead, the node values are directly evaluated by the JudgeAgent, significantly reducing simulation time.
>
> As shown in Figure 6 of the revised manuscript, compared with vanilla MCTS using only the InferAgent, the efficiency improvements brought by M²‑Miner grows exponentially with task complexity, achieving a 64× speedup at task length 9. Furthermore, the ablation results show that each agent is essential for boosting mining efficiency.

---

> ### Author Response · Authors · 2025-11-22
> **Response to Reviewer eLqw (Part 3/3)**
>
> **Q3: How the LLMs receive GUI state inputs and produce executable actions, and whether environment feedback is included in this loop?**
>
> We have added a detailed description of the infrastructure framework for the mining and training processes in Section 4.1 of the revised manuscript. And we have detailed the envoriment interaction in Appendix B.6 of the revised manuscript. We would like to address this concern in detail below.
>
> We use the Android Studio virtual machine (Android API~36) as our sandbox environment. We use adb commands to capture GUI screenshots from the virtual machine. To control the virtual machine, we parse the structured output (*i.e.,* JSON code) of the GUI Agent into executable adb commands. The feedback from the environment is included in the loop.
>
> In each iteration of the mining algorithm, we first obtain a screenshot of the virtual machine via adb commands. The Python process then retrieves the screenshot and combines it with a text prompt to construct the prompt. This prompt is fed to the inferAgent, which then produces a structured representation (*i.e.,* JSON code) of an action.
>
> We parse the structured representation into the corresponding adb command, send the command to the virtual machine, and the virtual machine executes the operation accordingly. This completes one cycle of state acquisition → action reasoning → operation execution.
>
> ---
>
> **References**
>
> [1] Zhang, Chi, et al. "Appagent: Multimodal agents as smartphone users." Proceedings of the 2025 CHI Conference on Human Factors in Computing Systems. 2025.
> [2] Li, Wei, et al. "On the effects of data scale on ui control agents." Advances in Neural Information Processing Systems 37 (2024): 92130-92154.
> [3] Zhang, Jiwen, et al. "Android in the zoo: Chain-of-action-thought for gui agents." Findings of the Association for Computational Linguistics: EMNLP 2024. 2024.
> [4] Lu, Quanfeng, et al. "GUIOdyssey: A Comprehensive Dataset for Cross-App GUI Navigation on Mobile Devices." Proceedings of the IEEE/CVF International Conference on Computer Vision. 2025.
> [5] Zhang, Zhong, et al. "AgentCPM-GUI: Building Mobile-Use Agents with Reinforcement Fine-Tuning." arXiv preprint arXiv:2506.01391 (2025).
> [6] Sun, Qiushi, et al. "Os-genesis: Automating gui agent trajectory construction via reverse task synthesis." Proceedings of the 63rd Annual Meeting of the Association for Computational Linguistics (Volume 1: Long Papers). 2025.
> [7] Huang, Zhiyuan, et al. "Spiritsight agent: Advanced gui agent with one look." Proceedings of the Computer Vision and Pattern Recognition Conference. 2025.
> [8] Ye, Jiabo, et al. "Mobile-agent-v3: Fundamental agents for gui automation." arXiv preprint arXiv:2508.15144 (2025).
> [9] Luo, Run, et al. "Gui-r1: A generalist r1-style vision-language action model for gui agents." arXiv preprint arXiv:2504.10458 (2025).
> [10] Liu, Yuhang, et al. "Infigui-r1: Advancing multimodal gui agents from reactive actors to deliberative reasoners." arXiv preprint arXiv:2504.14239 (2025).

---

> > ### Comment · Reviewer_eLqw · 2025-11-27
> >
> > Thanks. Most of my concerns are addressed. I will increase my rating.

---

> > > ### Author Response · Authors · 2025-11-27
> > > **Official Comment by Authors**
> > >
> > > Thank you for your active engagement and valuable contributions to improving our paper. We sincerely appreciate your support and endorsement!

---

### Official Review · Reviewer_WPrJ · 2025-11-01

**Soundness:** 3
**Presentation:** 3
**Contribution:** 3
**Rating:** 4
**Confidence:** 3

**Summary:**

This paper proposes M²-Miner, an automated framework for mining mobile GUI agent training data using Monte Carlo Tree Search (MCTS) enhanced with a collaborative multi-agent system. The framework introduces InferAgent, OrchestraAgent, and JudgeAgent to enhance expansion and simulation efficiency, an intent recycling strategy that extracts multiple intent-trajectory pairs from a single search tree, and a progressive model-in-the-loop training approach. Experiments show that GUI agents trained on M²-Miner data achieve state-of-the-art performance on several mobile GUI benchmarks while significantly reducing annotation costs.

**Strengths:**

The intent recycling strategy re-evaluates sibling paths to extract multiple intent-trajectory pairs from a single search tree, significantly improving data diversity and mining efficiency without additional exploration costs.

The progressive model-in-the-loop training implements a three-stage training strategy, allowing agent capabilities to improve progressively in tandem with data complexity, which enhances the mining success rate in unseen scenarios.

**Weaknesses:**

- The ablation study should be expanded: include a baseline using the stronger 72B model for InferAgent and JudgeAgent, but without the model-in-the-loop (MITL) strategy. This is necessary to validate the true effectiveness of MITL.

- The paper mentions using 8 A100-80G GPUs for training and "retraining for 2 epochs on the full mined dataset at each stage". These significant computational costs, as well as the API costs for Qwen2.5-VL-72B, seem to be omitted from the 196 total cost claimed in Table 1. It is better to clarify whether the 196 figure covers this computational overhead.

- The dataset partitioning is unclear. Your training set (during warm-up ) and test set both use AC and AITZ. You need to provide more detailed information, such as statistics on the exact splits, to clarify how data overlap is prevented, especially since ICLR appendices have no page limits.

**Questions:**

Did you build a custom framework to run the model-in-the-loop training strategy? If so, please provide more details. If not, please specify the base framework used.

---

> ### Author Response · Authors · 2025-11-22
> **Response to Reviewer WPrJ**
>
> Thank you for the detailed and constructive feedback! We treasure the opportunity to address your concerns and improve our work.
>
> ---
>
> **W1: Ablation study should be expanded: include stronger model without the model-in-the-loop strategy.**
>
> Following the suggestion, we have extended our model-in-the-loop (MITL) ablation study. To accelerate this experiment, we replace Qwen2.5-VL-72B with the recently open-sourced Qwen3-VL-30B-A3B model, which is noted for its superior performance and faster inference speed than Qwen2.5-VL-72B. The superiority of the Qwen3-VL-30B-A3B model over Qwen2.5-VL-72B, particularly on agent tasks, is documented on its official HuggingFace model card.
>
> Specifically, we utilize the Qwen3-VL-30B-A3B model for InferAgent and JudgeAgent without applying the MITL strategy. As shown in the table below, without MITL, Qwen3-VL-30B-A3B achieves limited gains on MSR and DQA compared to Qwen2.5-VL-7B, but still underperforms Qwen2.5-VL-7B with MITL. These results demonstrate the effectiveness of MITL in enhancing the framework's mining performance.
>
> | Method                          | MSR  | DQA  |
> |---------------------------------|:------:|:------:|
> | Qwen2.5-VL-7B w/o MITL           | 38.1 | 54.0 |
> | Qwen3-VL-30B-A3B w/o MITL        | 40.4 | 56.0 |
> | Qwen2.5-VL-7B w/ MITL            | 46.8 | 63.0 |
>
> This is attributed to the model's continuous learning of knowledge specific to the target applications. In contrast, while more powerful models possess superior general capabilities, they are still constrained by their limited understanding of new mobile applications.
>
> ---
>
> **W2: Cost calculation should include computational overhead.**
>
> We have added a detailed analysis of computational overhead in the **Cost Calculation and Comparison** section of our General Response.
>
> ---
>
> **W3: Statistics on the exact splits for utilized datasets should be more detailed.**
>
> In the table below, we provide detailed statistics on the composition of the datasets used. The division between the training and test sets strictly follows the official split to ensure consistency with the original data. These details have been incorporated into Appendix B.3 of the revised manuscript.
>
>
> | Dataset                                      | Platform | Average Steps | Total Trajectories | Train Split | Test Split |
> |:----------------------------------------------|:----------:|:--------:|:-------------:|:-------------:|:------------:|
> | AndroidControl              | Mobile   | 5.5    | 15,283      | 13,602      | 1,681      |
> | AITZ                     | Mobile   | 6.0    | 2,504       | 1,998       | 506        |
> | GUI Odyssey                | Mobile   | 15.3   | 7,735       | 5,801       | 1,934      |
> | AMEX                  | Mobile   | 12.8   | 2,946       | 2,946       | 0          |
> | CAGUI                    | Mobile   | 7.5    | 600         | 0           | 600        |
>
> ---
>
> **Q1: Did you build a custom framework to run the model-in-the-loop training strategy?**
>
> Yes, we build a customized infrastructure framework that supports mobile agent data mining and model-in-the-loop training. We have added a detailed description of this custom framework in Section 4.1 and Appendix B.7 of the revised manuscript. As illustrated in Figure 4 of the revised manuscript, our framework adopts a layered architecture, consisting of the data layer, engine layer, algorithm layer, agent layer, execution layer, and environment layer.
>
> The data layer organizes intents, interaction trajectories, screenshots, and metadata via an intent–trajectory tree structure. The engine layer implements a progressive training framework that starts with pre‑training and then goes through three subsequent stages (Stage 1 to Stage 3), enabling support for a model‑in‑the‑loop training strategy. The algorithm layer centers on a Monte Carlo Tree Search (MCTS) controller to handle selection, expansion, simulation, and backpropagation. The agent layer introduces multi-agent collaboration mechanisms for inference, orchestration, and reward evaluation during data mining. The execution layer integrates action parsing with toolchains, translating text/action parsing into ADB interface calls such as click, type, and swipe. The environment layer provides sandbox and virtual machine cluster support including environment control, isolation and security, and resource monitoring.
>
> Our layered framework unifies GUI agent execution, intent-trajectory data mining, and model training into a whole, thereby forming a customized model-in-the-loop framework that can automatically perform data mining and training.

---

### Author Response · Authors · 2025-11-22
**General Response**

We would like to firmly express our gratitude to all reviewers for their insightful and constructive comments, and appreciate the comments like *"model-in-the-loop training enhances the mining success rate in unseen scenarios"* (reviewer WPrJ), *"framework is scalable and adaptable, demonstrating potential for broader mobile applications"* (reviewer eLqw), *"intent recycling further enhances mining efficiency and intent richness"* (reviewer xyHC), and *"M$^2$-Miner reduces the time and cost of manual annotation; the framework demonstrates sufficient novelty"* (reviewer gnp5).

Our work focuses on the research of GUI Agent data mining. This is crucial for reducing the cost of manual annotation and for enhancing the model's execution performance on new mobile applications from a data–centric perspective. We pioneer the integration of Monte Carlo Tree Search into the mobile intent–trajectory data mining task and exponentially improve mining efficiency through a proposed collaborative multi–agent framework. We believe our work will make an impactful contribution and fill a notable gap in the production of intent–trajectory data within the Mobile GUI Agent domain.

Once again, we thank all reviewers for the time and effort invested in helping us refine our work! We would be eager to hear from you if there are any further concerns we have not yet addressed.

---

**We address a common concern below:**

**Cost Calculation and Comparison (Reviewer WPrJ and gnp5)**

We are sincerely grateful for the reviewers' constructive suggestions regarding the cost calculation, which helps us to further improve our work. In the revised manuscript, we provide a more detailed breakdown of the calculation process in Appendix B.6 and supplement it with the computational overhead. We provide a detailed description of the cost calculation below. All monetary values below are in US dollars ($).


**Cost for quality inspection.**

The 196 cost corresponding to our method in Table 1 is the cost for the manual quality inspection. It is calculated using the following formula:

\begin{equation}
\text{Cost} = (\text{Number of Images}) \times (\text{Quality Inspection Efficiency}) \times (\text{Hourly Wage Rate}),
\end{equation}

Plugging in the values:

\begin{equation}
196 = 20{,}000\ \text{images} \times 0.0014\ \text{hours/image} \times 7/\text{hour},
\end{equation}

The cost for other methods includes both annotation and quality inspection costs, calculated as follows:

\begin{equation}
\text{Cost} = (\text{Number of Images}) \times (\text{Annotation Efficiency} + \text{Quality Inspection Efficiency}) \times (\text{Hourly Wage Rate}),
\end{equation}

We perform manual quality verification by instructing quality inspectors to determine whether each mined trajectory correctly aligns with its associated intent. All values above are provided by a professional annotation team and can be found in Section 5.2 of our revised submission.


**Cost for computational overhead.**

We estimate the computational overhead based on the pricing of Vast.ai, a GPU rental platform. The total computational cost consists of two phases: training and data mining.

The training stage utilizes a rig with 8×A100–80G GPUs. For the model–in–the–loop strategy, each stage (including warm up) takes approximately 6 hours.  Thus, we require a 24–hour rental of this 8×A100–80G setup.

For data mining, we deploy the Qwen2.5–VL–7B and 72B models on a rig with 8×L40–45G GPUs. We run 32 concurrent mining processes, which yielded a total of 2,565 trajectory sequences. The process had an approximate 50% success rate (i.e., a 50% mining loss). With each trajectory taking about 10 minutes to mine, a 26.7–hour rental of the 8×L40–45G setup was necessary.

The rental fees are 0.924/hour for a single A100 and 0.433/hour for a single L40. The total computational cost is therefore calculated as:

- Training Cost: 24 hours × 8 GPUs × 0.924/GPU/hour = 177.4
- Mining Cost: 26.7 hours × 8 GPUs × 0.433/GPU/hour = 92.5
- Total Computational Cost: 177.4 + 92.5 = 269.9


**Total Cost.**

Therefore, with the addition of the 196 for quality inspection, the total cost amounts to 466. We have updated this value in Table 1. Compared to the similarly–sized AITZ dataset, our method reduces the construction cost by 6,010. When comparing on per–image cost (calculated as total cost divided by the number of images), our method is approximately 18 times more cost–effective than all other datasets.

---

### Meta-Review · Area_Chair_1beC · 2026-01-12

**Summary:**

This paper proposes M²-Miner, a data-centric framework for automated mining of intent–trajectory pairs for mobile GUI agents. The method instantiates a tree-search (MCTS) mining process enhanced by a collaborative multi-agent design (InferAgent for action proposal, OrchestraAgent for merging/prioritization, JudgeAgent for process-based reward estimation), plus intent recycling and progressive model-in-the-loop (MITL) training. Reviewers agreed the work is practically relevant (high-quality GUI trajectory data remains crucial) and the submission provides extensive experiments showing that training on mined data yields strong performance on multiple mobile GUI benchmarks while reducing reliance on manual annotation.

**Reviewer Concerns:**

The main reviewer concerns were (i) novelty (MCTS is classic; is the contribution mainly engineering?), (ii) clarity of the multi-agent interaction and ablation methodology, (iii) missing/dated baselines, (iv) cost accounting (omitted compute / API costs), and (v) dataset splits / leakage and evaluation rigor.

The rebuttal addressed the majority of these convincingly: it added targeted ablations (including stronger-model / no-MITL comparisons), clarified the role boundaries and information flow among agents during MCTS, expanded baseline coverage (including newer/concurrent methods), and provided a more complete cost breakdown including computational overhead. Concerns around split statistics and leakage were also directly addressed with additional documentation and clarification.

A remaining (but, in my view, non-fatal) concern is that the novelty is primarily compositional/systems-oriented rather than introducing a fundamentally new algorithmic primitive. However, the end-to-end framework, demonstrated mining efficiency gains, and broad experimental validation together constitute a solid contribution in a data-production direction that is still under-explored for mobile GUI agents.

**Reviewer Scores:**

- WPrJ (initially below threshold): Likely +1 if fully engaged post-rebuttal; requested ablations + cost accounting + split details were provided, but the reviewer did not re-engage after responses.
- eLqw: Would increase (and explicitly indicated so) after concerns about presentation/baselines/agent interaction were addressed.
- xyHC: Likely no change; maintained the score and noted limited domain familiarity.
- gnp5: Likely modest increase or no change; most detailed questions were clarified, though the reviewer remained focused on novelty relative to prior data-mining approaches.

---

### Decision · Program_Chairs · 2026-01-26

Accept (Poster)